# Investments in photoreceptors compete with investments in optics to determine eye design

**Francisco JH Heras, Simon B Laughlin***

Department of Zoology, University of Cambridge, Cambridge, United Kingdom

## eLife Assessment

This paper makes a **valuable** contribution to our understanding of the tradeoffs in eye design - specifically between improvements in optics and in photoreceptor performance. The authors successfully build a formal theory that enables comparisons across a wide range of species and eye types. One notable example is that how space should be allocated to optics and photoreceptors depends on eye type - with particularly notable differences between compound and simple eyes. The framework introduced to compare different design properties is **convincing** and provides a nice example of how to study tradeoffs in seemingly disparate design properties.

**\*For correspondence:**
SL104@cam.ac.uk

**Competing interest:** The authors declare that no competing interests exist.

**Abstract** Eyes provide opportunities to understand the function, design, development, and evolution of elaborate sense organs. We take a new cost–benefit approach to understanding eye design by considering that optics and photoreceptors compete for the resources invested in an integrated system. We investigate this competition theoretically and empirically using a new measure of cost, specific volume. This common currency for optics and photoreceptors relates investments to image quality via geometrical, optical, and physiological constraints. By covering the morphospace of an eye of given type and cost, we model how trading optics against photoreceptors changes information capacity. In apposition compound eyes and simple eyes, an optimum configuration maximises efficiency. Efficiency requires heavy investment in photoreceptors and depends on photoreceptor energy consumption. Optimum information capacities and efficiencies scale non-linearly with total investment. Diurnal insects' apposition eyes follow trends that promote efficiency: photoreceptor arrays take 40–80% of total specific volume, photoreceptor length increases systematically with spatial resolution, and photoreceptors are exceptionally long. Thus, competition between optics and photoreceptors shapes eye design, and matching investments in optics and photoreceptors to improve efficiency is a design principle. Our new methodology can be developed to view the adaptive radiation of eyes through a cost–benefit lens.

## Introduction

The design of eyes has long fascinated biologists with its variations, innovations, numerous adaptations of form to function, and insights into the evolution of organs, both rudimentary and perfectly contrived (*Darwin, 1859*; *Land and Fernald, 1992*; *Nilsson and Land, 2012*; *Oakley and Speiser, 2015*; *Walls, 1942*). We address a question of eye design that has not been considered before, but is familiar to people who construct imaging systems on tight budgets. How should the resources invested in an eye be divided between an eye's two major components, the optical system that forms images and the photoreceptor array that captures images?

**eLife digest** Animal eyes come in many shapes and sizes, from simple eye pits lined with light receptors to complex structures with bent lenses and corneas, as well as compound eyes. Nonetheless, the underlying principle is the same: an optical system forms an image that is captured by a photoreceptor array.

Over millions of years, eyes have adapted to an organism's lifestyle and habitat and can therefore vary not only between species but also between sexes and developmental stages. For example, fast-flying predators such as dragonflies (with compound eyes) and hawks (with simple eyes) have larger eyes with larger lenses, enabling them to resolve finer spatial detail. However, although larger eyes generally perform better, most eyes remain relatively small because they are energetically expensive organs. Thus, eye design reflects a balance between costs and benefits.

Previous studies of eye design have largely considered costs and benefits separately, overlooking an important interaction: optics and photoreceptors compete for limited resources. Greater investment in optics improves image quality and information projected onto photoreceptors, while greater investment in photoreceptors enhances the system's ability to capture that information. This raises a key question: is this competition a significant factor in eye design?

Heras and Laughlin addressed this question by introducing a mathematical framework that links investment directly to performance. Their approach allows modelling of how information capture changes as resources shift between optics and photoreceptors.

Using mathematical models of two types of compound eyes and one simple eye, they showed that all eye types studied possess an optimal allocation of resources that maximizes information capture. They further demonstrated that the energy costs of photoreceptors play a critical role in shaping an optimized eye. Applying their model to insect compound eyes, they estimated how resources are distributed between optics and photoreceptors and found that, as predicted, insects invest efficiently by allocating more resources to photoreceptors than to optics. Such an alignment of investments to maximize efficiency thus represents a new principle of eye design.

Research on how sensory systems are adapted to an organism's lifestyle and habitat has largely focused on benefits rather than costs. The method developed by Heras and Laughlin provides a powerful tool for understanding how cost-benefit relationships have shaped the evolution of eyes, brains and behavior. Further progress will depend on improved measurements of costs. Their work also suggests that the remarkable energy efficiency of brains may arise from a similar balancing of cost–benefit functions across their component systems.

---

The benefits of enlarging the dioptric apparatus and the photoreceptor array are well understood; consequently, numerous studies relate measures of eye morphology (radius, width, length, lens diameter, pupil area, and photoreceptor length) to optical, geometrical, and physiological constraints on performance to show how morphology adapts an eye to lifestyle and habitat (*Barlow, 1952*; *Kirschfeld, 1976*; *Land, 1981*; *Nilsson and Land, 2012*; *Niven et al., 2007*; *Snyder, 1979*; *Labhart and Nilsson, 1995*; *Land, 1997*; *Thomas et al., 2020*). However, the fact that an eye's performance depends on two systems that effectively compete for the resources invested in an eye, the dioptric apparatus and the photoreceptor array, is rarely considered (*Kirschfeld, 1976*; *Kröger and Biehlmaier, 2009*). We investigate how trade-offs between dioptrics and photoreceptor array determine an eye's performance and efficiency by developing a new cost–benefit approach, which uses measures of eye morphology to calculate costs, and relates these measures to optical, geometrical, and physiological constraints to calculate benefits.

We consider three basic eye types, two types of apposition compound eye, neural superposition (NS) and fused rhabdom, and a simple eye (*Figure 1*). We establish and road test our cost–benefit approach using apposition compound eyes because they offer a unique combination of advantages. The optical and physiological constraints are well documented, their effects on performance are measured and modelled (*Snyder, 1979*; *Stavenga, 2004b*; *Howard et al., 1987*; *Hardie and Postma, 2008*; *Song et al., 2012*; *Heras and Laughlin, 2017*) and their influence on eye design is well established. For example, the compound eyes of houseflies, praying mantises, dragonflies, and robber flies have foveas for detecting, locating, and catching flying mates or prey. Here, the facet lens

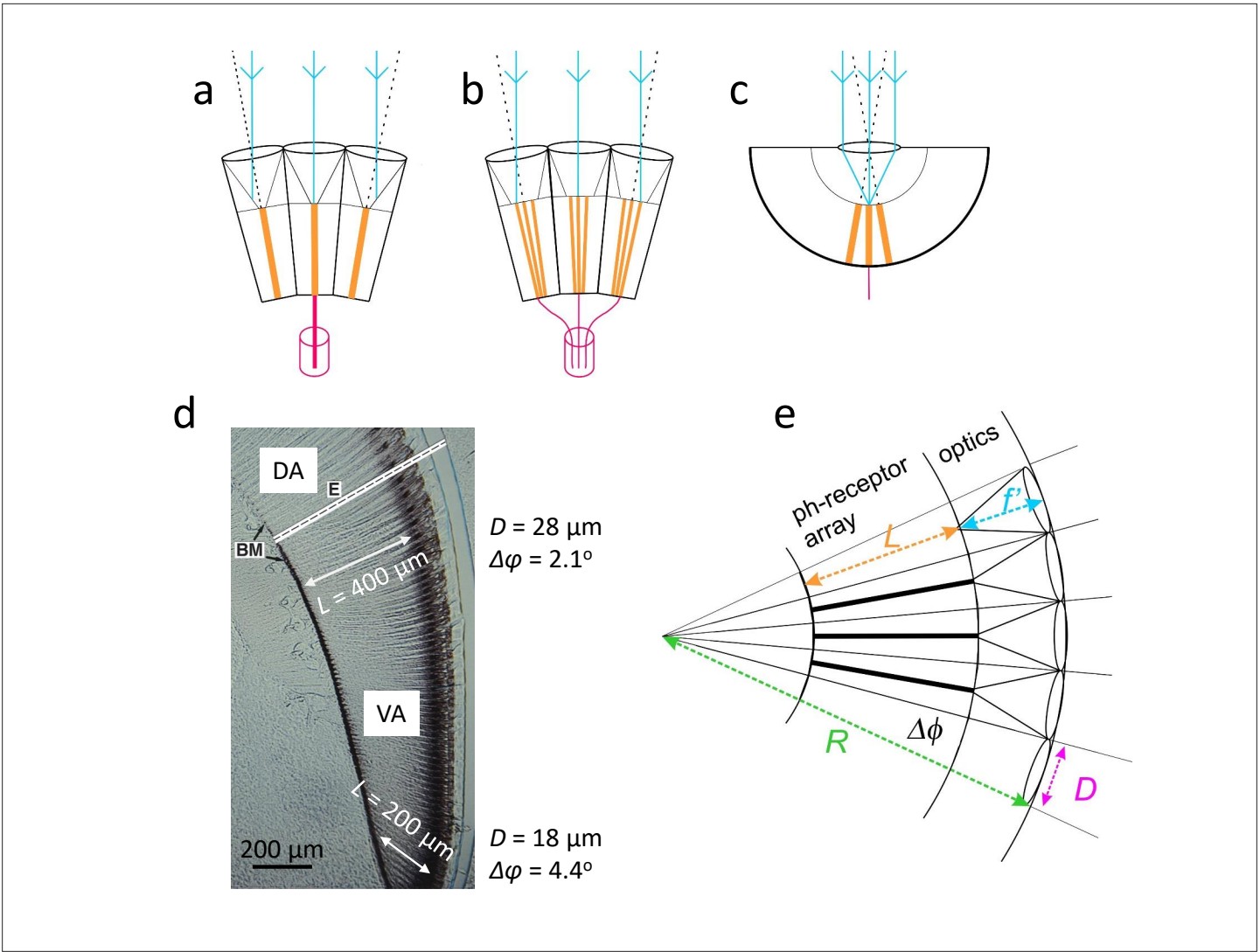

**Figure 1.** Eye structure and geometry define resolution and costs. (**a**) Fused rhabdom apposition eye, photoreceptors coding a pixel form fused rhabdom and send axons to a single neural module; (**b**) neural superposition (NS) apposition eye, photoreceptor forms its own rhabdomere, photoreceptors with same optical axes code single pixel and send axons to single neural module; and (**c**) simple eye, as in a camera each photoreceptor codes a single pixel. (**d**) Gradient of investment in spatial acuity: apposition eye, honeybee drone, *Apis mellifera*. From ventral to dorsal, lens diameter $D$ increases and interommatidial angle $\Delta\phi$ decreases to increase spatial resolution, and rhabdom length, $L$, increases to increase $SNR_{ph}$. 10 µm thick longitudinal section, (DA) dorsal eye area and (VA) ventral area. BM – retina's basement membrane; E – equator separating dorsal and ventral regions. From *Menzel et al., 1991*, original micrograph courtesy of Doekele Stavenga. (**e**) Schematic section of locally spherical apposition eye region. Volumes of optics and photoreceptor array are determined by dimensions that constrain the quality of the spatial image coded by photoreceptors: lens diameter $D$, focal distance $f'$, interommatidial angle $\Delta\phi = D/R$ where $R$ is eye radius, and rhabdom(ere) length $L$.

is enlarged to reduce the diffraction limit to optical resolving power, the ommatidia are more densely packed to increase angular resolution, and photoreceptors are elongated to increase signal to noise ratio by transducing more photons (*Rossel, 1979*; *Hardie, 1985*; *Labhart and Nilsson, 1995*; *Burton and Laughlin, 2003*; *Wardill et al., 2017*). In addition, there are clear indications that photoreceptors are costly and investments in optics and photoreceptors are matched. Apposition eyes' photoreceptor arrays obviously fill a substantial fraction of eye volume because photoreceptors construct long photoreceptive waveguides (rhabdomeres, rhabdoms), substantial photoreceptor energy costs increase with rhabdomere length (*Niven et al., 2007*), and there is a precise, fine scale covariation of acuity and photoreceptor length, measured in two apposition eyes (*Labhart and Nilsson, 1995*; *Rossel, 1979*) and obvious in many others (e.g. *Figure 1d*).

Our new cost–benefit approach establishes that photoreceptor cost is a major factor in eye design that influences optics by competing for the resources invested in an eye. Consequently, matching investments in optics and photoreceptors to improve efficiency is a principle of eye design. This principle can explain why apposition eyes have longer photoreceptors than simple eyes and photoreceptor length increases with spatial acuity in apposition and simple eyes. We suggest that our approach can advance our understanding of visual ecology by costing the benefits of the adaptations observed in eyes (*Cronin et al., 2014*; *Nilsson and Land, 2012*). Moreover, because the allocation of resources to optics and photoreceptors both promotes cost-effectiveness and depends upon developmental and physiological mechanisms (*Casares and McGregor, 2021*; *Niven and Laughlin, 2008*), our cost–benefit approach may well contribute to our understanding of how and why eyes evolve (*Nilsson and Pelger, 1994*).

## Results

We start with the neural superposition (NS) eyes of flies. This type of apposition compound eye provides both the detailed analyses of optical and physiological constraints required to develop a model that relates investments in optics and photoreceptor arrays to performance, and the published data on dioptrics and photoreceptor arrays required to test our model. Moreover, because fly NS eyes vary in size between species while retaining similar structure and function, we simplify our study by developing and using a generic fly NS model. We then model and examine fused rhabdom apposition eyes to identify general principles of apposition eye design. Finally, we model a simple (camera) eye to demonstrate that changing eye type changes the distribution of costs. In all three types, we consider diurnal eyes operating in daylight when the number of available photons is not limiting. Although this restriction eliminates the many ways in which eyes are designed to operate efficiently at lower light levels (*Nilsson and Land, 2012*), it makes it easier to establish points of principle by reducing the numbers of variables in our models. The symbols we use for variables and parameters are listed in *Table 1*.

### Calculating the benefits and costs of investing in optics

In daylight, when the number of available photons is not limiting, investments in optics buy increased optical contrast and spatial resolving power by reducing optical and geometrical constraints, as follows. An apposition eye forms and captures images with an array of ommatidia (*Figure 1*). Each ommatidium uses a facet lens of diameter, $D$, and focal length (in air), $f$, to focus light onto the entrance aperture of photoreceptive waveguides, a single fused rhabdom or several rhabdomeres (*Figure 1*) of diameter $d_{rh}$. Increasing $D$ sharpens a photoreceptor's angular sensitivity by reducing the blur produced by the Airy disk diffraction pattern, whose half-width $\Delta\rho_l = \lambda/D$ radians, where $\lambda$ is the wavelength of light, taken to be 500 nm. Increasing $f$ and reducing $d_{rh}$ also sharpen angular sensitivity by reducing the angular subtense of the photoreceptor entrance aperture, $d_{rh}/f$. Sharpening angular sensitivity has two benefits: it increases spatial resolving power and increases image contrast.

We calculate the effects of $D$, $d_{rh}$, and $f$ on the half-width of a fly photoreceptor's angular sensitivity, $\Delta\rho$, using two formulae, Snyder's simplification, his convolution of Gaussians (CoG) $\Delta\rho^2 = \Delta\rho_l^2 + \Delta\rho_{rh}^2$, where $\Delta\rho_l = \lambda/D$ and $\Delta\rho_{rh} = d_{rh}/f$; and Stavenga's better approximation, $\Delta\rho = 1.26\Delta\rho_l$, obtained using a comprehensive wave-optics model (WOM) of flies' NS eyes operating in bright light (see Methods for more details). For CoG, we fix $d_{rh} = 1.9$ µm because this value is typical of fly R1–6 (*Hardie, 1985*). CoG provides a lower bound to optical resolving power and WOM a more efficient upper bound that, in bright light, makes better use of investments in optics by operating closer to the diffraction limit.

The benefits of investing in optics come at a cost: increasing $D$ and lengthening $f$ to reduce $\Delta\rho$ expands the volume of the dioptric apparatus, $V_o$, which in apposition eyes is easily calculated from eye geometry (*Figure 1e*). $D$ and $\Delta\phi$ define the radius $R = D/\Delta\phi$ of a locally spherical eye region, in which the dioptric apparatus (lens, cone, and screening pigment) is a shell of thickness $f'$, the focal distance from lens front surface to focal plane. Thus,

$$V_o = \frac{1}{3}[R^3 - (R - f')^3]\,\mu m^3\,sr^{-1}. \tag{1}$$

**Table 1.** Symbols used in the Results and the Discussion.

| Symbol | Description | Units |
| --- | --- | --- |
| $D$ | Lens diameter | μm |
| $f$ | Lens focal length, in air | μm |
| $f'$ | Focal distance | μm |
| $F$ | F-number | |
| $d_{rh}$ | Diameter of rhabdom (fused-rhabdom eye) or rhabdomere (NS eye) | μm |
| $\Delta\phi$ | Interommatidial or interreceptor angle | rad (equations) |
| | | ° (figures) |
| $R$ | Radius of eye or locally spherical eye region | μm |
| $p$ | Eye parameter | |
| $L$ | Length of rhabdom or rhabdomere, also depth of photoreceptor array | μm |
| $\lambda$ | Wavelength of light | nm |
| $\eta_i$ | Refractive index of eye's internal medium | |
| $\Delta\rho_l$ | Half width of lens point-spread function | rad |
| $\Delta\rho_{rh}$ | Half-width of rhabdom(ere) acceptance angle | rad |
| $\Delta\rho_{ph}$ | Half-width of photoreceptor acceptance angle | rad |
| $V_o$ | Volume of optics | μm³ sr⁻¹ |
| $V_{ph}$ | Volume of photoreceptor array | μm³ sr⁻¹ |
| $V_{tot}$ | Total eye volume | μm³ sr⁻¹ |
| $C_o$ | Cost of optics | μm³ sr⁻¹ |
| $C_{ph}$ | Costs of photoreceptor array | μm³ sr⁻¹ |
| $C_{tot}$ | Total cost of eye | μm³ sr⁻¹ |
| $S_E$ | Photoreceptor energy surcharge | μm³ sr⁻¹ |
| $K_E$ | Photoreceptor energy tariff | μm³ per microvillus |
| $N_{vil}$ | Number of microvilli | |
| $\nu$ | Number of microvilli per unit length of rhabdom or rhabdomere | μm⁻¹ |
| $\psi$ | Transduction rate | s⁻¹ |
| $SNR_{ph}$ | Photoreceptor signal to noise ratio | per unit contrast |
| $H$ | Spatio-temporal information capacity | bits sr⁻¹ s⁻¹ |

Because $V_o$ is a specific measure, volume per unit solid angle of visual space, we can compare investments in eye regions that differ in radius and angular extent. Thus, $V_o$ handles the regional variations in $R$ observed in many compound eyes.

To simplify our expression for $V_o$ (**Equation 1**) we fix lens F-number, $F = f/D$ and apply the formula for image formation by a convex cornea, $f' = f.n_i$, where $n_i$ is the refractive index of the internal medium (**Nilsson and Land, 2012**). This formula describes several apposition eyes, including fly NS eyes (**Rossel, 1979**; **Stavenga et al., 1990**; **Stavenga, 2003a**; **Zeil, 1983**). Thus,

$$V_o = \frac{1}{3}[R^3 - (R - Fn_iD)^3]\,\mu m^3\,sr^{-1}. \tag{2}$$

We use values measured in blowfly, $F = 2.0$ (**Stavenga, 2003a**) and $n_i, = 1.34$ (**Seitz, 1968**).

Fixing $F$ both simplifies our models by reducing the number of free parameters and accounts for the dependence of the photon flux entering an ommatidium on $D$ by making image brightness (the flux per unit area of rhabdomere entrance aperture) independent of $D$. We use the specific volume of optics, $V_o$, as our measure of the cost of optics, $C_o$ μm³ sr⁻¹. This usage assumes that the cost of materials, the cost of metabolic energy for maintaining function, and the energy cost of carriage all increase in equal proportion to volume, in all parts of the dioptric apparatus, namely the corneal lens, cone, and surrounding pigment cells. Given the lack of definitive data on composition and costs, this assumption is a reasonable starting point (Discussion).

## Calculating the benefits and costs of investing in photoreceptors

Investing in photoreceptors buys photoreceptor signal-to-noise ratio, $SNR_{ph}$, by reducing the effects of an optical constraint, the Poisson statistics of photon absorption (**Nilsson and Land, 2012**), and at higher light levels a physiological constraint, the saturation of a photoreceptor's transduction units – its light sensitive microvilli (**Heras and Laughlin, 2017**; **Howard et al., 1987**; **Song et al., 2012**). A microvillus contains rhodopsin molecules, the intermediate molecules of the phototransduction cascade and the ion channels they activate, and it uses these to generate a brief pulse of depolarising current, a quantum bump, following the activation of a single rhodopsin molecule by an absorbed photon. This all-or-none response takes time to complete, during which the microvillus cannot respond to the absorption of another photon (**Hardie and Postma, 2008**; **Song and Juusola, 2014**). Therefore, as light intensity rises to daylight levels, $SNR_{ph}$ falls below the Poisson limit because a photoreceptor's transduction units saturate (**Howard et al., 1987**). At any given time, a significant fraction of microvilli are failing to respond to the absorption of a photon by rhodopsin because they are already engaged in processing a bump. Consequently, the transduction rate $\psi$ no longer increases linearly with the absorption rate, and bump statistics change from Poisson to binomial. In daylight, flies activate a photomechanical response, a longitudinal pupil, to prevent excessive saturation and maintain $SNR_{ph}$ close to the maximum permitted by binomial statistics,

$$SNR_{ph} = \frac{1}{2}\sqrt{N_{vil}}, \tag{3}$$

where $SNR_{ph}$ is defined with respect to a signal of unit contrast (**Howard et al., 1987**). Note that by operating at this upper limit, the fly obtains maximum benefit from investing in $N_{vil}$. Thus, the benefit of investing in a longer rhabdomere is an increase in $SNR_{ph}$ in bright light, according to

$$SNR_{ph} = \frac{1}{2}\sqrt{\nu L}, \tag{4}$$

where $L$ is rhabdomere length and $\nu$ is the number of microvilli per unit length. We assume $\nu$ is constant along the rhabdomere and from published results we estimate $\nu = 230$ μm⁻¹ (Methods). We acknowledge that diameter, cross section, and taper often vary within and among rhabdomeres. We are fixing $\nu$ at a plausible value for a generic model.

The cost of increasing $SNR_{ph}$ by increasing $L$ is easily calculated. In fly NS eyes, $L$ is also the depth of the photoreceptor array because the rhabdomeres of the six largest photoreceptors, R1–6, stretch from the focal plane of the lens to the basement membrane (**Hardie, 1985**). Thus in a locally spherical eye region (**Figure 1**) the photoreceptor array's specific volume is

$$V_{ph} = \frac{1}{3}[(R - f')^3 - (R - f' - L)^3]\, \text{μm}^3\, \text{sr}^{-1}. \tag{5}$$

Substituting $f' = F.n_i.D$

$$V_{ph} = \frac{1}{3}[(R - Fn_iD)^3 - (R - Fn_iD - L)^3]\, \text{μm}^3\, \text{sr}^{-1}. \tag{6}$$

According to eye geometry (**Figure 1**), the total specific volume of an eye region is

$$V_{tot} = V_o + V_{ph}\, \text{μm}^3\, \text{sr}^{-1}. \tag{7}$$

Observe that because both $V_o$ (*Equations 1 and 2*) and $V_{ph}$ (*Equation 5; Equation 6*) depend upon $R$ and $R = D/\Delta\phi$, both $V_o$ and $V_{ph}$ increase with the number of ommatidia per unit solid angle.

For the photoreceptor array, we assume that the costs of space, materials and energy for carriage increase in proportion to volume with the same constants of proportionality as optics. Again, in the absence of measurements, this assumption is a reasonable starting point (Discussion). However, this assumption breaks down when we consider specific metabolic rates. To enable and power phototransduction, a photoreceptor has an exceptionally high specific metabolic rate (energy consumed per gram, and hence unit volume, per second) (*Laughlin et al., 1998*; *Niven et al., 2007*; *Pangršič et al., 2005*). We account for this extra cost by applying an energy surcharge, $S_E$. To equate with our other estimates of cost, this surcharge has units of $\mu m^3\, sr^{-1}$, and we assume that it increases in proportion to $N_{vil}$ because $N_{vil}$ determines the magnitude of the photoreceptor's light-gated conductance (*Hardie and Postma, 2008*; *Heras and Laughlin, 2017*). We satisfy these two requirements by defining

$$S_E = K_E N_{vil}\, \mu m^3\, sr^{-1}, \tag{8}$$

where $K_E$, the photoreceptor energy tariff, converts the energy consumed by a photoreceptor per microvillus into an equivalent volume.

We estimate $K_E$ (Methods) by adopting the method used to estimate the energy cost of weapons carried by flying beetles (*Goyens et al., 2015*). We divide the energy consumed per microvillus by the animal's mass specific metabolic rate to obtain an equivalent body mass which, assuming a density of 1.0, is our volume equivalent, $K_E$ per microvillus. $K_E$ is at present poorly determined: it depends upon several ecological, physiological and behavioural factors, it will vary among species and among individuals of the same species, and there is barely enough data (Methods). Given this uncertainty, we model a range of values from $K_E = 0$ to $K_E = 0.64$ that covers the range we estimate for blowfly, $K_E = 0.13$ to $K_E = 0.52$ (Methods).

To summarise costs, for the photoreceptor array

$$C_{ph} = V_{ph} + S_E\, \mu m^3\, sr^{-1}, \tag{9}$$

and for the optics

$$C_o = V_o\, \mu m^3\, sr^{-1}, \tag{10}$$

Because $C_o$ and $C_{ph}$ have the same units and are specific (per unit solid angle), we can transfer resources between optics and photoreceptors within the constraint of total specific cost

$$C_{tot} = C_o + C_{ph}\, \mu m^3\, sr^{-1}. \tag{11}$$

## Modelling the effects of resource allocation on NS eye performance

When $C_{tot}$ is fixed, allocating a smaller proportion to optics and a larger proportion to photoreceptors, trades the benefits of investing in optics for the benefits of investing in photoreceptors (*Figure 2*).

Having used constraints to link investments in dioptrics and photoreceptor array to three major determinants of the quality of the achromatic images captured by fly photoreceptors R1–6, namely $\Delta\rho$, $\Delta\phi$, and $SNR_{ph}$, we can now model how trade-offs between optics and photoreceptors change an eye's ability to support vision. Several measures of visual performance depend on $\Delta\rho$, $\Delta\phi$, and $SNR_{ph}$. To establish proof of principle, we use a general measure that embraces a wide variety of resolvable achromatic image details – spatio-temporal information capacity in bits $sr^{-1}\, s^{-1}$. Information capacity has proved useful for measuring the performance of compound eyes (*de Ruyter van Steveninck and Laughlin, 1996*), discovering design principles (*van Hateren, 1992*; *Howard and Snyder, 1983*; *Snyder et al., 1977*), and illuminating the evolution of simple eyes (*Nilsson and Pelger, 1994*). The measure is particularly relevant for a fly NS eye because coding by second order neurons is adapted to maximise information capacity (*van Hateren, 1992*; *Laughlin, 1981*).

To model the effects of resource allocation on information capacity, we fix $C_{tot}$ and generate pairs of $D$ and $L$ that cover the range of values permitted by eye geometry. Each $D, L$ pair specifies an investment in optics, $C_o(D, L)$, an investment in photoreceptors $C_{ph}(D, L)$ and hence the three determinants of image quality, $\Delta\rho(D, L)$, $\Delta\phi(D, L)$, and $SNR_{ph}(D, L)$, that specify spatio-temporal information capacity $H(D, L)$ bits $sr^{-1}\, s^{-1}$.

We calculate $H(D, L)$ (Methods) in the frequency domain (*Howard and Snyder, 1983*; *van Hateren, 1992*). In brief, a 2D power spectrum typical of natural scenes is low-pass filtered by the photoreceptor angular sensitivity function, $\Delta\rho(D, L)$, and sampled by a hexagonal lattice of photoreceptors specified by $\Delta\phi(D, L)$. Movement of the retinal image modulates the flux of photons entering a photoreceptor by converting spatial frequencies into temporal frequencies, according to the distribution of angular velocities generated during behaviour. Then, during transduction, the photoreceptors low-pass filter these temporal frequencies and add the noise generated by sampling the photon flux stochastically with transduction units (microvilli). We use (*van Hateren, 1992*) formulae for the $1/f^2$ power spectrum of natural scenes and the distribution of image velocities appropriate for blowfly, and we temporally low-pass filter according to the measured properties of a fully light-adapted blowfly photoreceptor R1–6 (Methods). We also modify van Hateren's method to take account of the spatial aliasing that occurs when an array of ommatidia undersamples an image (Methods).

By calculating information capacities for each $D, L$ pair, we define the performance surface $H(D, L)$ that covers the morphospace of model eyes of given $F$-number, $F$, and specific cost, $C_{tot}$. The performance surface is a flat-topped ridge with steep flanks (*Figure 3*). Atop the ridge, a single point of maximum capacity, $H(D_{opt}, L_{opt})$, sits in an extensive zone within which capacity, and hence efficiency, is > 95% maximum (red zone in *Figure 3*). With efficiency dropping steeply away from the ridge, approximately two thirds of $D, L$ combinations have efficiencies < 70%.

The shape of the performance surface depends upon photoreceptor energy cost (*Figure 3*). Increasing the energy tariff $K_E$ reduces $L_{opt}$ from the maximum permitted by eye geometry to one quarter of maximum. However, because the ridge is flat-topped, $L_{opt}$ still lies in an extensive high-efficiency zone, within which $L$ can be changed threefold and $D$ by 40%, while maintaining > 95% efficiency (*Figure 3*). This broad high-efficiency zone is bad news for investigators who set out to establish that eyes are optimised, but good news for insects. An eye region can be adapted for a specific purpose, for example by increasing $L$ to increase $SNR$, without sacrificing more than 5% of a

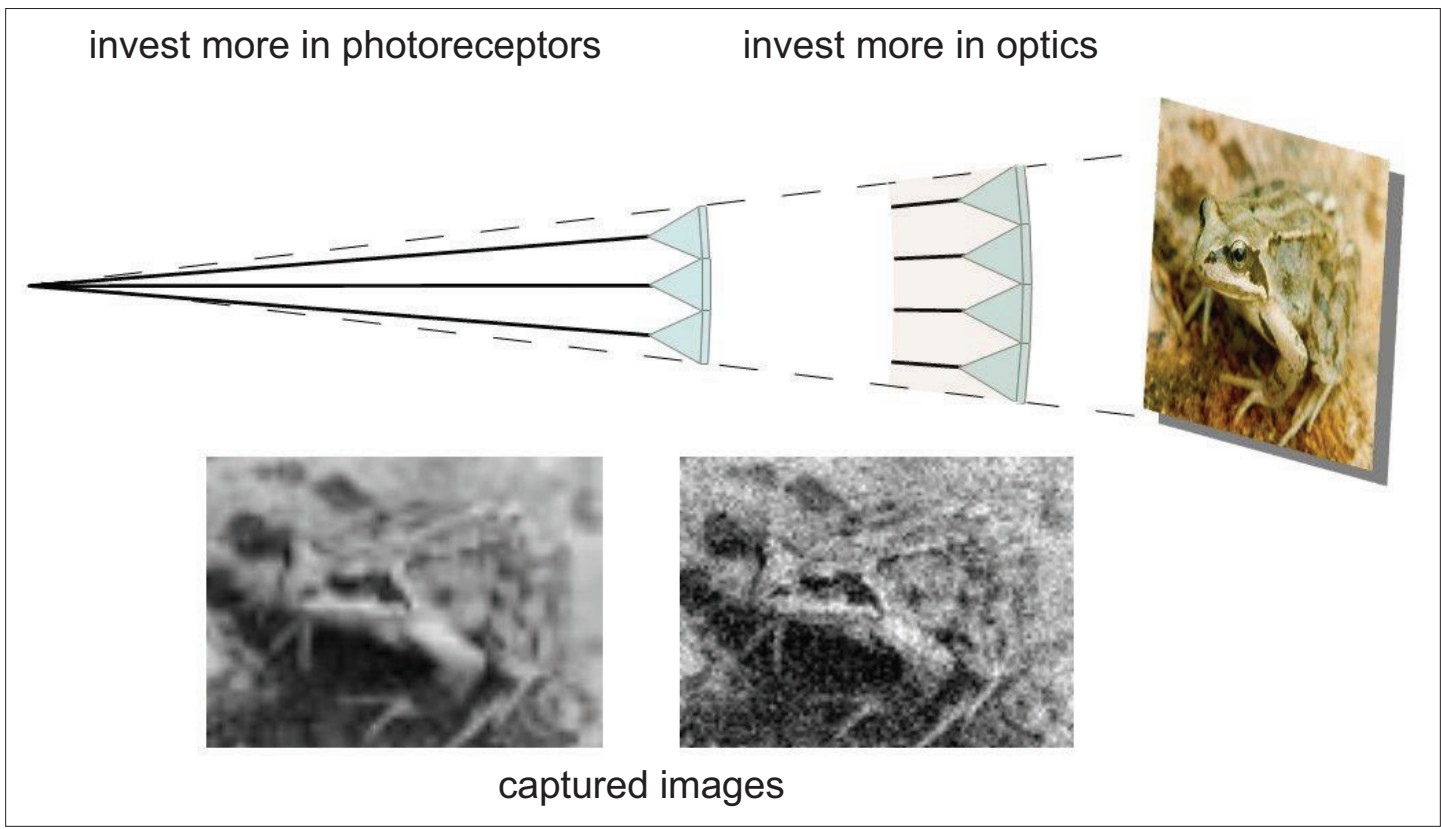

**Figure 2.** The effect of trading investment in photoreceptor array for investment in dioptric apparatus. The schematic shows two eye regions of equal specific volume, the left investing more heavily in photoreceptors, the right more heavily on optics. The images they capture show that transferring resources from photoreceptor array to optics increases image sharpness and contrast at the expense of increasing noise.

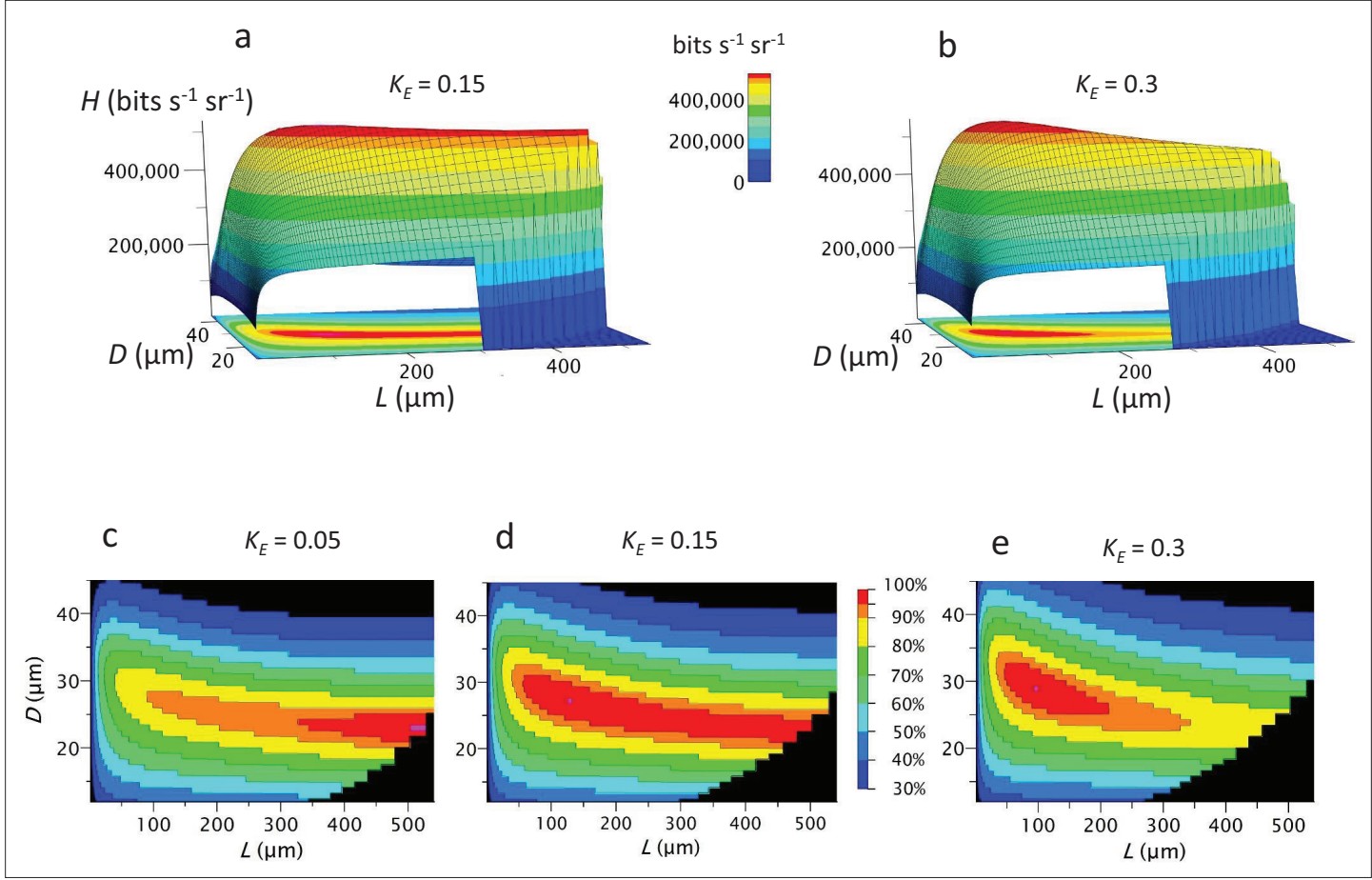

**Figure 3.** The performance surface $H(D, L)$ describes how information capacity changes across all geometrically permissible configurations of an eye of fixed cost – the eye's morphospace. Red area: high-performance zone in which capacity is >95% maximum. Surfaces plotted for model fly neural superposition (NS) eyes, with F-number, $F = 2$ and total cost, $C_{tot} = 4 \times 10^9$ μm³ sr⁻¹ at three values of photoreceptor energy tariff $K_E$. Acceptance angle $\Delta\rho$ calculated using **Snyder, 1979** CoG approximation. Data for all graphs is provided in **Figure 3—source data 1**. The Matlab code used to calculate performance surfaces across the morphospaces of the model eyes studied in this paper is publicly available at https://github.com/fjhheras/eyedesign, copy archived at **Heras, 2025**.

The online version of this article includes the following source data for figure 3:

**Source data 1.** Related to **Figure 3a–e**.

measure of general-purpose vision, information capacity. Thus, this robust region of design space can facilitate evolution by reducing loss of function.

## Patterns of investment in optimised NS eye models

We next establish how, in theory, eye structure and investments in optics and photoreceptor array vary with total investment when efficiency is optimised by maximising information capacity. We run our fly NS eye model with different combinations of $C_{tot}$ and $K_E$ and for each combination find the values of $L_{opt}$, $D_{opt}$, and $\Delta\phi_{opt}$ that maximise information capacity and the corresponding investments in optics and photoreceptor array, $C_o$ and $C_{ph}$. For each combination of $C_{tot}$ and $K_E$ we run two versions of our NS model, one calculates $\Delta\rho$ using CoG (**Snyder, 1979**) and the other WOM, which produces a narrower $\Delta\rho$ that comes closer to the lens diffraction limit (**Stavenga, 2004a**). The two versions give similar patterns of investment (**Figure 4**).

### Rhabdomere length is matched to acuity and rhabdomeres are long

As $C_{tot}$ increases, $D_{opt}$ widens to increase lens contrast transfer and resolving power by reducing $\Delta\rho$, $\Delta\phi_{opt}$ narrows to increase spatial resolution, and $L_{opt}$ lengthens to improve $SNR_{ph}$, reaching several

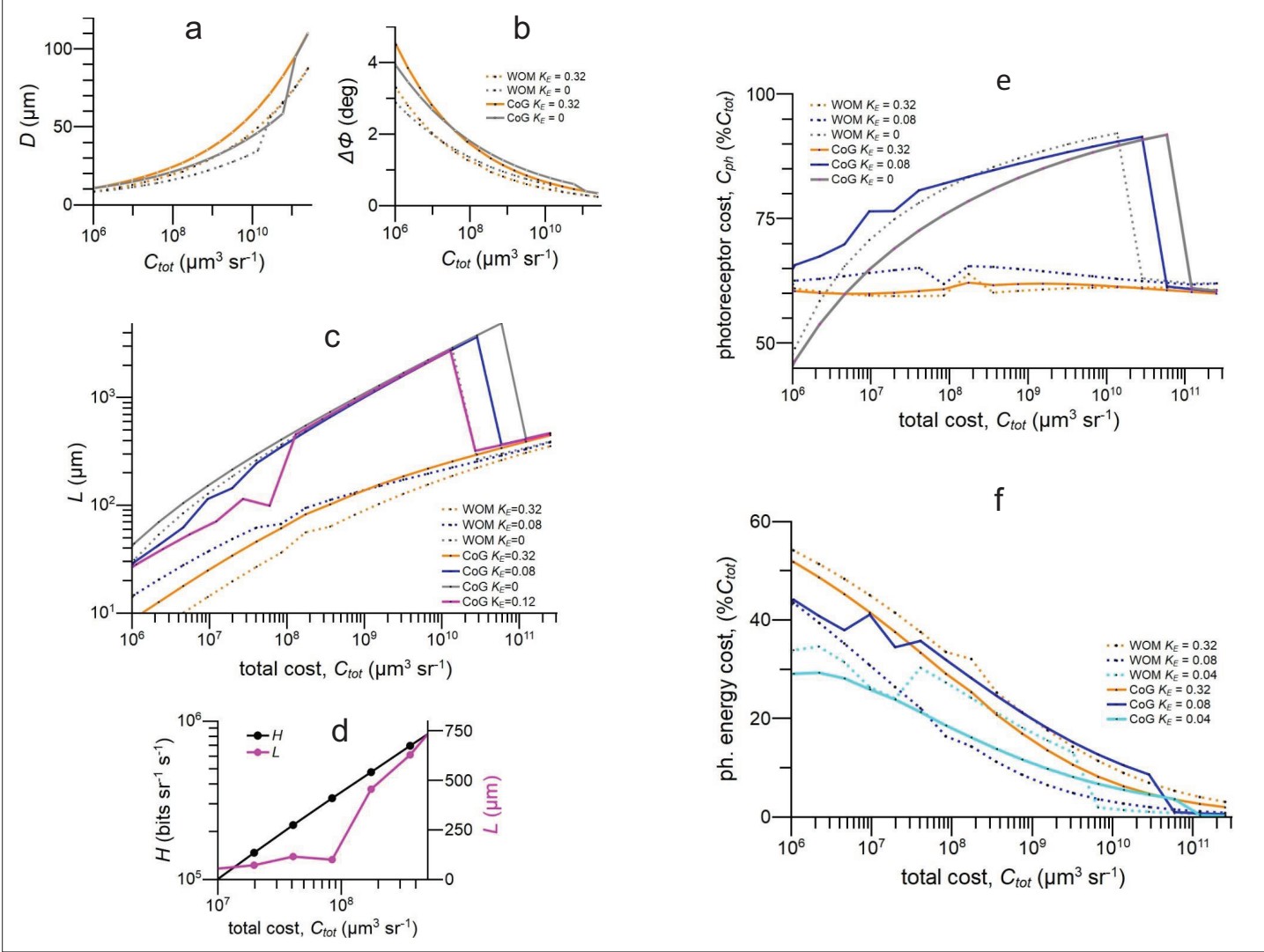

**Figure 4.** When model fly NS eyes are optimised for information capacity, eye structure, eye performance, and the division of resources between optics and photoreceptor array depend on total investment, $C_{tot}$ and photoreceptor energy tariff, $K_E$. (**a**) Lens diameter $D$; (**b**) interommatidial angle, $\Delta\phi$; and (**c**) rhabdomere length, $L$. Note the sudden jump in the curve for CoG $K_E = 0.12$. (**d**) Optimum information capacity $H$ and rhabdomere length $L$ at CoG $K_E = 0.12$ plotted over the range of $C_{tot}$ within which $L$ jumps upward. The eightfold increase in $L$ has very little effect on the rate of increase of $H$. (**e**) %$C_{tot}$ allocated to photoreceptor array. (**f**) Photoreceptor energy cost as %$C_{tot}$. CoG, acceptance angle approximated by convolving Gaussians (**Snyder, 1979**); WOM, acceptance angle approximated according to wave optics model (**Stavenga, 2004a**). Key in (**b**) also applies to (**a**). Data for all graphs is provided in **Figure 4—source data 1**.

The online version of this article includes the following source data for figure 4:

**Source data 1.** Related to **Figure 4a–f**.

hundred microns in larger eyes (**Figure 4a–c**). Thus, efficient resource allocation can, in theory, explain why NS eye rhabdomeres are long and rhabdomere length increases with spatial acuity. Note that improving the performance of the optical system by approaching the lens diffraction limit to resolving power improves an eye's performance in full daylight. With the WOM approximation for $\Delta\rho$ the lens diameter $D$ required to achieve a given $\Delta\rho$ is decreased, and this allows $\Delta\phi$ to decrease (**Figure 4b**), thereby increasing the spatial resolution achieved by an efficient eye of given total cost.

## Photoreceptor energy cost changes the structure of an efficient eye

Increasing the photoreceptor energy tariff $K_E$ has little effect on the optimum values of $D$ and $\Delta\phi$, but it reduces the optimum value of $L$, and hence the depth of the photoreceptor array, by as much as 90%

(*Figure 4c*). The optimum value of $L$ is most sensitive to $K_E$ in the range 0.04–0.16 and tends to jump up and down as $C_{tot}$ increases. Careful interrogation of our MATLAB code suggests that coding and rounding errors are not responsible for these jumps. We attribute the jumps to interactions between competing non-linear terms (e.g. the dependence of noise on $L^{0.5}$, information on log($SNR$), effective spatial bandwidth on $R^{0.5}$, volume on $R^3$) that shift the optimum across a flat-topped performance surface. This explanation is supported by the observation that these large jumps in optimum $L$ have very little effect on the optimum information rate (*Figure 4d*). The effects of $K_E$ on eye structure are also sensitive to optical performance: increasing $K_E$ has a greater effect on $L_{opt}$ in the more sharply focussed WOM model at all but the highest values of $C_{tot}$ (*Figure 4c*). As expected, increasing $K_E$ reduces the spatial resolution achieved at a given $C_{tot}$ by reducing the resources available for investment in the volume costs $V_o$ and $V_{ph}$.

## Photoreceptor arrays are allocated more resources than optics

When our NS eye models are optimised for information capacity, more than 50% of $C_{tot}$ is allocated to photoreceptor arrays over the range $C_{tot} = 2 \times 10^6$ to $3 \times 10^{11}$ µm³ sr⁻¹, irrespective of optical effectiveness (CoG cf. WOM) and photoreceptor energy tariff, $K_E$ (*Figure 4e*). At lower values of $K_E$, the optimum allocation to photoreceptor arrays is strongly dependent on $C_{tot}$, increasing from 45% to 90% $C_{tot}$ over the range $C_{tot} = 1 \times 10^6$ µm³ sr⁻¹ to $1 \times 10^{10}$ µm³ sr⁻¹, over the range and then jumping down to approximately 60% between $C_{tot} = 1 \times 10^{10}$ µm³ sr⁻¹ and $C_{tot} = 1 \times 10^{11}$ µm³ sr⁻¹ (*Figure 4e*). At higher values of $K_E$, the optimum photoreceptor allocation is relatively insensitive to $C_{tot}$ and lies between 60% $C_{tot}$ and 65% $C_{tot}$. We conclude that, in theory, the cost of the photoreceptor array is as significant as the cost of optics. Therefore, to fully understand eye design, both costs must be taken into account.

The relative contributions of photoreceptor array volume, $V_{ph}$, and photoreceptor energy cost, $S_E$, to total array cost, $C_{ph}$, depends on total investment, $C_{tot}$ (*Figure 4f*). At our lowest $C_{tot}$, that is in the lowest acuity eyes regions, the energy cost is 25–50% of $C_{tot}$, depending on $K_E$, but as eye size increases the costs associated with array volume come to dominate and photoreceptor energy cost drops to 5% of $C_{tot}$.

**Table 2.** Measurements of Dipteran neural superposition (NS) eyes, extracted from published sources (*Land, 1997*; *Hardie, 1985*; *Stavenga et al., 1990*; *Stavenga, 2003a*; *Land and Eckert, 1985*; *Stavenga, 2003b*; *Gonzalez-Bellido et al., 2011*; *Zeil, 1983*; *Wardill et al., 2017*).
Further details of measurements made, their use and results obtained are given in *Supplementary file 1*.

| Species | Eye region | $D$ (µm) | $\Delta\phi$ (°) | $L$ (µm) | $f$ (µm) | $f'$ (µm) | $F$ | Source |
|---|---|---|---|---|---|---|---|---|
| *Calliphora vicina* | Male acute zone | 37 | 1.07 | 340 | 74 | 99 | 2 | *Land, 1997*; *Hardie, 1985*; *Stavenga et al., 1990*; *Stavenga, 2003a* |
| *Calliphora vicina* | Female acute zone | 29 | 1.28 | 280 | 58 | 78 | 2 | *Hardie, 1985* |
| *Musca domestica* | Male love spot | 36 | 1.75 | 250 | 72 | 96 | 2 | *Hardie, 1985*; *Land and Eckert, 1985* |
| *Musca domestica* | Male peripheral | 20 | 3.5 | 140 | 40 | 54 | 2 | *Hardie, 1985* |
| *Drosophila melanogaster* | Average | 16.5 | 5 | 83 | 20 | 27 | 1.25 | *Land, 1997*; *Hardie, 1985*; *Stavenga, 2003b* |
| *Coenesia attenuata* | Female frontal | 20 | 2.3 | 140 | 25 | 28 | 1.25 | *Gonzalez-Bellido et al., 2011* |
| *Bibio marci* | Male dorsal | 33 | 1.6 | 230 | 70 | 94 | 2.1 | *Zeil, 1983* |
| *Bibio marci* | Male ventral and female | 21 | 3.7 | 110 | 36 | 48 | 1.7 | *Zeil, 1983* |
| *Dilophus febrilis* | Male dorsal | 24 | 2.2 | 180 | 40 | 54 | 1.7 | *Zeil, 1983* |
| *Dilophus febrilis* | Male ventral and female | 15 | 5.1 | 60 | 16 | 21 | 1.1 | *Zeil, 1983* |
| *Holcocephela fusca* | Acute zone, max. acuity | 75 | 0.28 | 230 | 160 | 176 | 2.53 | *Wardill et al., 2017* |
| *Holcocephela fusca* | Peripheral | 21 | 3.5 | 123 | 33 | 45 | 1.6 | *Wardill et al., 2017* |

## Flies invest efficiently in costly photoreceptor arrays

Our new method for calculating costs enables us to estimate, for the first time, the total investment an animal makes in an eye or eye region, and break this down into its two major components, optics and photoreceptor array. Using published measurements of $D$, $\Delta\phi$, and $L$ we determine the specific volumes of dioptric apparatus, $V_o$, and photoreceptor array, $V_{ph}$ in fly NS eyes. Although numerous publications give values of $D$ and $\Delta\phi$ (**Land, 1997**; **Feller et al., 2021**), very few include $L$. We have values for just 12 eye regions, taken from 7 species of flies (**Table 2**). For each region, we insert the values, $L$, $D$, $\Delta\phi$, and $f'$, into **Equations 1 and 5** to calculate $V_o$, $V_{ph}$, and $V_{tot}$. Where focal distance $f'$ is not reported (**Table 2**), we follow our models (**Equation 2**) and calculate $f' = FDn_i$ with $F = 2$ and $n_i = 1.34$.

The 12 eye regions provide a well-distributed set of empirical values of $L$, $D$, $\Delta\phi$, $V_o$, $V_{ph}$, and $V_{tot}$ that spans almost five orders of magnitude of $V_{tot}$, from $1.4 \times 10^6$ µm³ sr⁻¹ and $2.1 \times 10^6$ µm³ sr⁻¹for the low acuity eye region of the March fly *Dilophus febrilis* and the eye of $9 \times 10^{10}$ µm³ sr⁻¹ *Drosophila*

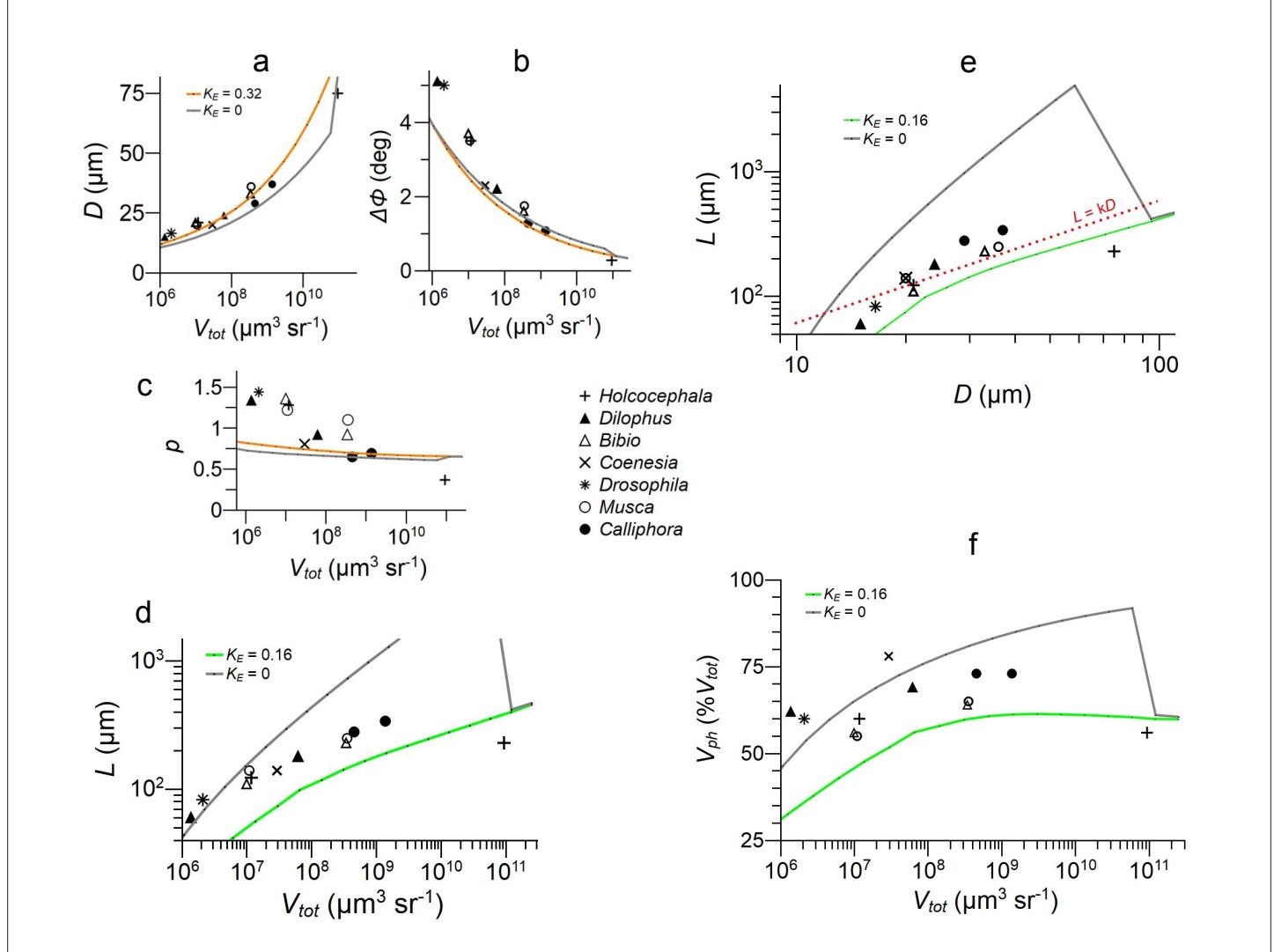

**Figure 5.** Parameters defining the configurations of 12 fly NS eye regions, taken from 7 species (**Table 2**), compared to model NS eyes optimised for information capacity at given values of photoreceptor energy tariff $K_E$, and total specific volume $V_{tot}$. (a) Lens diameter $D$, (b) interommatidial angle $\Delta\phi$, (c) eye parameter $p = D\Delta\phi$ , and (d) rhabdomere length $L$. (e) Log–log plot of $L$ vs $D$, dashed line shows slope for isomorphic scaling, $L \propto D$. (f) Specific volume of photoreceptor array, $V_{ph}$ expressed as % total specific volume of eye, %$V_{tot}$. Models use Snyder's CoG approximation for photoreceptor acceptance angle (**Snyder, 1979**). Data for all graphs is provided in **Figure 5—source data 1**.

The online version of this article includes the following source data for figure 5:

**Source data 1.** Related to **Figure 5a–f**.

*melanogaster*, respectively, to for the remarkable fovea of the small robber fly *Holcocephala fusca* (*Wardill et al., 2017*).

We compare the empirical values obtained from flies with theoretical curves, obtained from model NS eyes that are optimised for information capacity (*Figure 5*). Note that for our theoretical curves, we plot $D$, $\Delta\phi$, $L$, and $V_{ph}$ against total specific cost, $C_{tot}$ (*Figure 4*). This total includes the surcharge for photoreceptor energy consumption $S_E$, which increases in proportion to the number of microvilli and the photoreceptor energy tariff, $K_E$ (*Equation 8*). However, we cannot estimate $C_{tot}$ in flies because their values of $K_E$ are not well-determined (Methods). Although $K_E$ will vary considerably between species, we only have enough published data to calculate its order of magnitude and likely range (Methods). Furthermore, $K_E$ is species-specific and, because we have no more than two data points per species and we are using a generic model that does not account for differences in $d_{rh}$ and $F$ among species (*Table 2*), we cannot estimate $K_E$ by fitting our model to the empirical data. For these reasons, we compare theoretical findings to empirical data by plotting them against a measure of eye size that is both calculated by our models and estimated from data, total specific volume, $V_{tot}$. Because $K_E$ is uncertain, we plot theoretical curves calculated with values of $K_E$ from within our likely range.

Looking across 12 fly eye regions, $D$ increases and $\Delta\phi$ decreases as $V_{tot}$ increases. $D$ lies within, or close to, the theoretical envelope defined by CoG models optimised with $K_E = 0$ and $K_E = 0.32$ (*Figure 5a*), and tends to slightly exceed optimum values at lower specific volumes. $\Delta\phi$ is approximately 25% wider than optimum (*Figure 5b*) in the four eye regions with lowest specific volume and close to optimum elsewhere. The eye parameter, $p = D\Delta\phi$ (*Figure 5c*) indicates the degree to which an apposition compound eye sacrifices spatial resolution by sampling below the lens diffraction limit. Only *Holcocephala* fovea, $p = 0.37$, comes close to the diffraction limit $p = 0.29$ (*Snyder, 1979*). As observed in a number of diurnal apposition eyes (*Wehner, 1981*), the remaining 11 fly eye regions in our data set undersample by a factor of 2 or more, with $p$ decreasing with increasing $V_{tot}$ from 1.4 to 0.65 (*Figure 5c*). Our optimised models also undersample, with $p$ decreasing with increasing $V_{tot}$ from 0.89 to 0.65 (*Figure 5c*). Note that like $D$ and $\Delta\phi$, $p$ is relatively insensitive to $K_E$. We conclude that in most eye regions the eye parameter $p$ is not optimised for spatio-temporal information capacity, at least according to our model, because it is higher than predicted at all but the highest values of $V_{tot}$.

Rhabdomere length, $L$, increases with $V_{tot}$ from 60 μm to 340 μm (*Figure 5d*; *Table 2*), within or close to the theoretical envelope defined by $K_E = 0$ and $K_E = 0.16$. There is a robust trend: the empirical values of log $L$ increase with log $V_{tot}$, and hence with increasing $D$ and decreasing $\Delta\phi$ (*Figure 5a, b*), along a trajectory that resembles theoretical curves for optimum log $L$. However, the data do not follow the theoretical curve for any particular value of $K_E$, and there is some scatter among the data points (*Figure 5d*). We have three reasons to attribute most of this scatter to unknown differences in $K_E$. First, the optimum value of $L$ is sensitive to $K_E$ (*Figure 4c*); second, $K_E$ will vary between species (Methods); and third, the empirical values of $D$ and $\Delta\phi$, which are in theory relatively insensitive to $K_E$, are less scattered and lie closer to our theoretical predictions (*Figure 5a, b*). The one obvious outlier, the shorter than predicted rhabdomeres in *Holcocephala* fovea (*Figure 5d*), is easily explained: the optics and photoreceptor array of this forward pointing fovea, together with underlying optic lobe, are tightly packed into a head that is less than 600 μm from front to back (*Wardill et al., 2017*), and this lack of headroom limits the depth of the eye, $(L + f')$.

The log–log plot of $L$ vs $D$ (*Figure 5e*) shows that the increase in $L$ with $V_{tot}$ is not explained by proportional scaling because the empirical values do not lie on a straight line of slope 1, $L = kD$. As expected of a physiological system subject to several competing constraints (*Taylor and Thomas, 2014*), the theoretical curve continuously changes slope, and this is reflected in the data. Thus, the matching of $L$ to $D$ and $\Delta\phi$ observed in the data is not the consequence of a simple developmental constraint; it is the product of developmental mechanisms that allocate resources efficiently to optics and photoreceptor array.

We conclude that the eye regions we analyse are, according to our generic model, not optimised for spatio-temporal information capacity because the data points that are least sensitive to the photoreceptor energy tariff $K_E$, namely $D$, $\Delta\phi$ and $p$, are generally larger than predicted. Nonetheless, we argue on two grounds that the matching of rhabdomere length to spatial resolution seen in flies promotes the efficiency of their total investments in eye. First, $L$ lies within, or close to, the theoretical envelope predicted by models optimised for efficiency and second, because the theoretical

**Table 3.** Measurements of fused-rhabdom apposition eyes, extracted from published sources (*Rossel, 1979*; *Labhart and Nilsson, 1995*; *Menzel et al., 1991*; *Varela and Wiitanen, 1970*; *Kelber and Somanathan, 2019*).
Further details of measurements made, their use and results obtained are given in *Supplementary file 1*.

| Species | Eye region | $D$ (µm) | $\Delta\phi$ (°) | $L$ (µm) | $f$ (µm) | $f'$ (µm) | $F$ | Source |
|---|---|---|---|---|---|---|---|---|
| *Tenodora australasiae* | Eye coordinates $z = 15$; $x = 20$ (fovea) | 47.5 | 0.64 | 500 | 205 | 445 | 8 | *Rossel, 1979* |
| | $z = 15$; $x = 30$ | 51.5 | 0.8 | 535 | 335 | 405 | 6.5 | *Rossel, 1979* |
| | $z = 15$; $x = 10$ | 41 | 1 | 415 | 305 | 375 | 7.4 | *Rossel, 1979* |
| | $z = 15$; $x = 0$ | 35 | 1.1 | 285 | 205 | 275 | 5.9 | *Rossel, 1979* |
| | $z = 15$; $x = 60$ | 42 | 1.6 | 390 | 150 | 212 | 3.6 | *Rossel, 1979* |
| | $z = 15$; $x = 100$ | 33 | 2.5 | 330 | 110 | 153 | 3.3 | *Rossel, 1979* |
| *Sympetrum* spp. | Ventral –60° | 31 | 1.85 | 461 | 124 | 166 | 4 | *Labhart and Nilsson, 1995* |
| | Ventral –5° | 27 | 1.75 | 461 | 108 | 148 | | *Labhart and Nilsson, 1995* |
| | Dorso-ventral border | 31 | 1.69 | 516 | 124 | 167 | | *Labhart and Nilsson, 1995* |
| | Dorsal +50 | 53 | 1.82 | 627 | 92 | 123 | | *Labhart and Nilsson, 1995* |
| | Dorsal +70 | 61.5 | 1.78 | 886 | 246 | 330 | 4 | *Labhart and Nilsson, 1995* |
| | Dorsal fovea | 71 | 0.35 | 1107 | 305 | 410 | 4.4 | *Labhart and Nilsson, 1995* |
| *Apis mellifera* drone | Dorsal eye | 40 | 1 | 500 | 205 | 275 | 5.1 | *Menzel et al., 1991* |
| | Top ventral | 28 | 2.1 | 400 | 145 | 194 | 5.1 | *Menzel et al., 1991* |
| | Mid ventral | 25 | 2.8 | 280 | 82 | 110 | 3.3 | *Menzel et al., 1991* |
| | Basal ventral | 18 | 4.4 | 150 | 54 | 72 | 3 | *Menzel et al., 1991* |
| *Apis mellifera* worker | Max. acuity | 25 | 1.6 | 350 | 100 | 134 | 5.6 | *Varela and Wiitanen, 1970*; *Kelber and Somanathan, 2019* |

performance surface has a wide flat high-efficiency zone (*Figure 3*), large deviations from optimum have relatively little impact on efficiency.

To accommodate photoreceptors' lengthy rhabdomeres, flies allocate between 56% and 78% of eye volume, $V_{tot}$, to photoreceptor arrays (*Figure 5f*). The smaller remainder, between 44% and 22%, is allocated to optics. There is a slight trend for the percentage allocated to photoreceptors to increase with $V_{tot}$. With the exception of *Holcocephala*'s fovea, where the depth of the photoreceptor array is much less than predicted for optimum allocation, the volumes allocated to photoreceptor arrays lie within or close to the theoretical envelope defined by $K_E = 0$ and $K_E = 0.16$. On this basis, we conclude that our diurnal flies allocate a larger fraction of eye volume to photoreceptors than to optics, and this allocation promotes the efficient use of resources. Thus, photoreceptor costs are playing a major role in determining the design and efficiency of flies' NS eyes.

## Fused rhabdom apposition eyes show similar patterns of investment

We investigate diurnal apposition eyes with fused rhabdoms by modifying our NS eye model and comparing theoretical results with empirical data using the procedures developed for NS eyes. The modifications are straightforward. We increase lens $F$-number to 5.5 and reduce $d_{rh}$ to 1.8µm. These values are in the middle of the wide range found in our sample of apposition eye regions (*Table 3*). We also adjust the relationship between $L$ and the number of microvilli $N_{vil}$ to account for a fused rhabdom (Methods). For simplicity, we assume that all photoreceptors contribute equal numbers of microvilli along the full length of the rhabdom so that $N_{vil}$ increases in proportion to the depth of the photoreceptor array.

We then compare the models' predictions of the optimum values of $L, D, \Delta\phi, V_o, V_{ph}$, and $V_{tot}$ with empirical values extracted from published measurements of $D, \Delta\phi, L$, and $f'$ (*Table 3*). The necessary data is only available from studies of four species: the mantid *Tenodora australis* (*Rossel, 1979*), the honeybee *Apis mellifera* (*Kelber and Somanathan, 2019*; *Menzel et al., 1991*; *Varela and Wiitanen,*

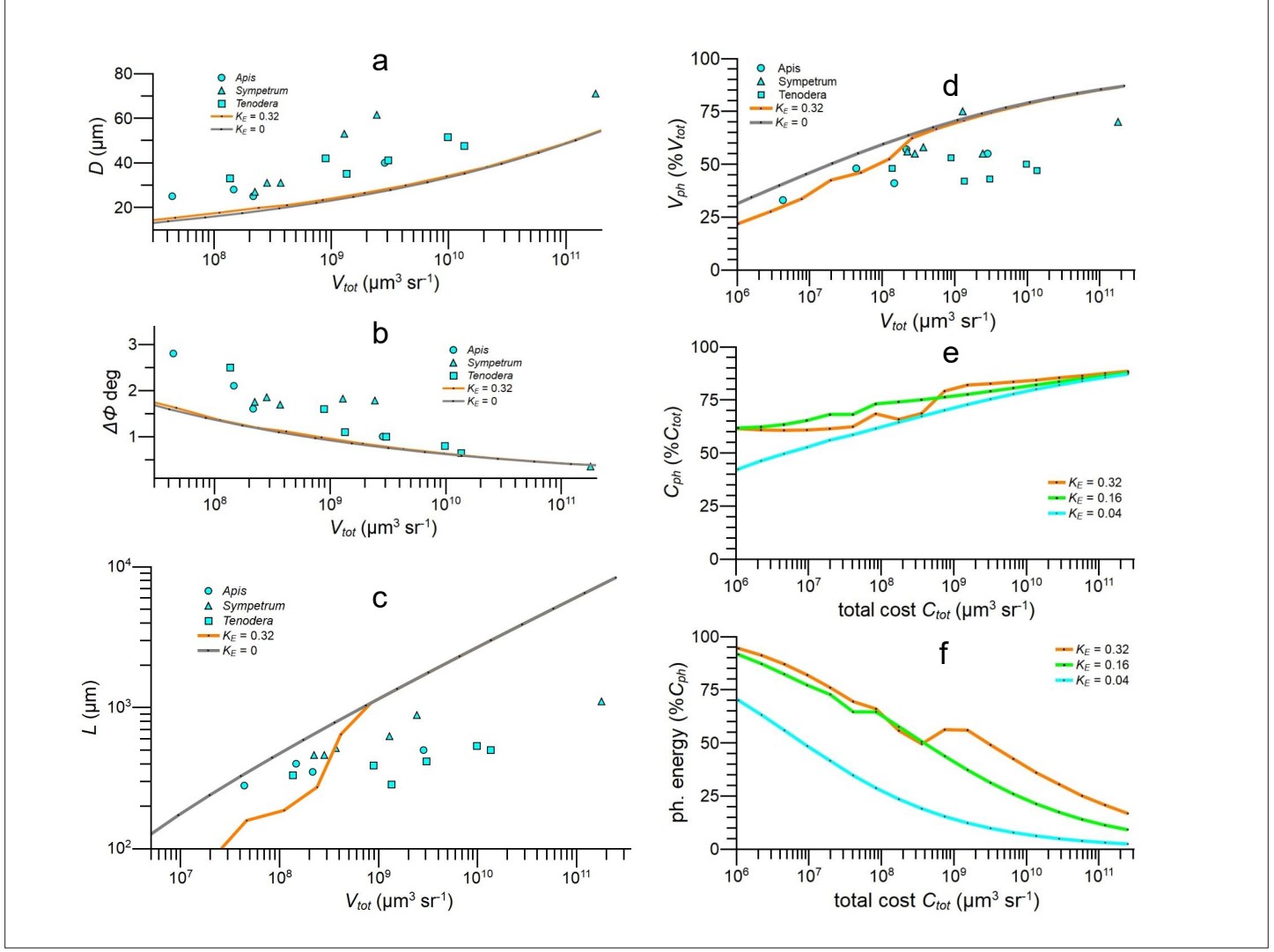

**Figure 6.** Theoretical results from apposition eye models optimised for information capacity compared to empirical data from 16 apposition eye regions taken from 3 species (*Table 3*). (**a**) $D$ vs specific volume $V_{tot}$. (**b**) $\Delta\phi$ vs $V_{tot}$. (**c**) $L$ vs $V_{tot}$. (**d**) photoreceptor array specific volume $V_{ph}$ as $\%V_{tot}$ vs $V_{tot}$. (**e**, **f**) Models demonstrate impact of photoreceptor costs and its dependence on total cost, $C_{tot}$. (**e**) Photoreceptor cost, $C_{ph}$, as $\%C_{tot}$. (**f**) Photoreceptor energy cost as $\%C_{ph}$. Models run with $F$-number $F = 5.5$, using COG approximation for acceptance angle (*Snyder, 1979*) and values of energy tariff $K_E$ given in keys. Data for all graphs is provided in *Figure 6—source data 1*.

The online version of this article includes the following source data for figure 6:

**Source data 1.** Related to *Figure 6a–f*.

1970), and two species of the Libellulid dragonfly *Sympetrum* that are so similar that their data are lumped together (*Labhart and Nilsson, 1995*). The studies provide empirical values for 16 eye regions, which span a range of $V_{tot}$, from $4.4 \times 10^7$ µm³ sr⁻¹ for *Apis* worker ventral eye to $1.8 \times 10^{11}$ µm³ sr⁻¹ for *Sympetrum* dorsal fovea (*Figure 6*).

Looking across our limited sample of fused rhabdom apposition eyes, $D$ increases with $V_{tot}$, $\Delta\phi$ decreases, and $L$ increases from 280 µm to 1100 µm (*Figure 6a–c*). These empirical values are more widely scattered than those for NS eyes (*Figure 5a, b, d*), and the discrepancies between empirical and predicted values are larger. We expect a wider scatter because our sample of fused rhabdom eyes is more diverse than our sample of NS eyes, both structurally and phylogenetically. In addition, the scatter is dominated by regional differences within eyes (*Figure 6*), several of which adapt a region for a specific task. For example, the 1100 µm rhabdomeres of the dragonfly *Sympetrum*, which are the longest photoreceptive waveguides found in any eye, adapt the fovea

for detecting small prey against a bright blue sky by enhancing $SNR_{ph}$ (**Labhart and Nilsson, 1995**). Despite the scatter in data points, fused rhabdom eyes follow the trends seen in NS eyes (**Figure 6** cf. **Figure 5**). In most eye regions, $D$ and $\Delta\phi$ are larger than predicted by models optimised for efficiency (**Figure 6a, b**), so therefore is the eye parameter $p$ (values not plotted). At low $V_{tot}$, $L$ lies within the theoretical envelope defined by models run with plausible values of photoreceptor energy tariff, $K_E$. However, at higher specific volumes $L$ falls increasingly short (**Figure 6c**). As with flies' NS eyes, large investments are made in photoreceptor arrays. The volume fractions of photoreceptor arrays increase from 33% $V_{tot}$ to 75% $V_{tot}$ as $V_{tot}$ increases, and the majority of values lie between 40% $V_{tot}$ and 60% $V_{tot}$ (**Figure 6d**). Our information maximisation models predict an increase from 33% $V_{tot}$ to 85% $V_{tot}$.

Our models also give the relative contribution of photoreceptor cost to total cost when fused rhabdom apposition eyes are optimised for spatial information capacity. The percentage of total investment devoted to the photoreceptor array increases with the energy tariff $K_E$ and climbs steadily with increasing $C_{tot}$, from 43% to more than 80% (**Figure 6e**). As in NS eyes, the contribution of photoreceptor energy consumption to array cost is progressively marginalised by array volume cost as $C_{tot}$ increases, falling to less than 20% in the largest eyes (**Figure 6f**).

We conclude that the 16 eye regions in our sample of fused rhabdom apposition eyes are not optimised for spatio-temporal information capacity, at least according to our models. However, the regions exhibit three trends that are associated with efficient resource allocation – rhabdomeres are relatively long, length increases with acuity, and the photoreceptor array occupies a large fraction of eye volume. These trends suggest that, as in NS eyes, resources are being divided between optics and photoreceptor array to promote efficiency. Consequently, photoreceptor costs play a major role in the design of fused-rhabdom apposition eyes.

## Patterns of efficient investment are different in simple eyes

To further investigate how investment patterns differ according to eye type, we construct a basic hemispherical model of a simple eye with rhabdomeric photoreceptors (Methods; **Figure 7a**) that captures the equivalence of $D$, $f'$, and $\Delta\phi$ in apposition and simple eyes (**Kirschfeld, 1976**). We then follow the procedures described above to see how optimum patterns of investment depend on total investment, $C_{tot}$, and photoreceptor energy tariff, $K_E$. We consider the same 5 orders of magnitude of total investment, $C_{tot} = 1 \times 10^6$ to $2.5 \times 10^{11}$ µm³ sr⁻¹, and for simple eye models this encompasses a 100-fold range of lens diameters, $D$ = 29 µm to 2.9 mm, and focal distances, $f'$ = 79 µm to 7.9 mm.

For each $C_{tot}$, there is an optimum division of resources between optics and photoreceptors that maximises information capacity. $L_{opt}$ increases with $V_{tot}$, scaling close to $(C_{tot})^{1/3}$ (**Figure 7b**). Increasing the energy tariff, $K_E$, greatly reduces $L_{opt}$ without changing this scaling exponent. At any given $C_{tot}$ and $K_E$, $L_{opt}$ is much shorter in simple eyes and the difference increases with $C_{tot}$ from at least 50% shorter at $C_{tot} = 1 \times 10^6$ µm³ sr⁻¹ to over 90% shorter at $C_{tot} = 1 \times 10^9$ µm³ sr⁻¹. We conclude that, in theory, efficient resource allocation explains why photoreceptors in diurnal simple eyes have much shorter light-sensitive waveguides than photoreceptors in diurnal apposition eyes.

Patterns of investment differ from apposition eyes in other respects (**Figure 7**). In simple models, the optimum volume fraction of the photoreceptor array is insensitive to $C_{tot}$ whereas in apposition models, it increases with $C_{tot}$. In addition, a simple eye is much more sensitive to photoreceptor energy consumption. The small energy tariff, $K_E = 0.04$, reduces simple eye array volume from 35% $V_{tot}$ to 12% $V_{tot}$, but has a negligible effect on an NS eye. Raising the tariff to $K_E = 0.32$, reduces the simple eye's array volume to just 3% $V_{tot}$ but in apposition models it is always > 30% $V_{tot}$. Despite this tenfold reduction in %$V_{tot}$ the total resources allocated to photoreceptor array increases from 35% to 57% (**Figure 7d**) because the relative contribution of photoreceptor energy consumption increases from 40% $C_{tot}$ to 55% $C_{tot}$. Finally, in simple eyes, the relative contribution of photoreceptor energy consumption is insensitive to $C_{tot}$, but in apposition models, it falls steadily with increasing $C_{tot}$, to around 5% $C_{tot}$ (**Figure 7d**). This difference between eye types can be attributed to differences in eye geometry (**Figure 7a**). As $C_{tot}$ increases, a simple eye's photoreceptors maintain their dense packing, but an apposition eye's photoreceptors become increasingly widely spaced as larger lenses force the boundaries between ommatidia further apart. We conclude that when resources are efficiently allocated, the differences in geometry that define eye type can profoundly influence both the impact of photoreceptor energy consumption on eye morphology and the distribution of costs within an eye.

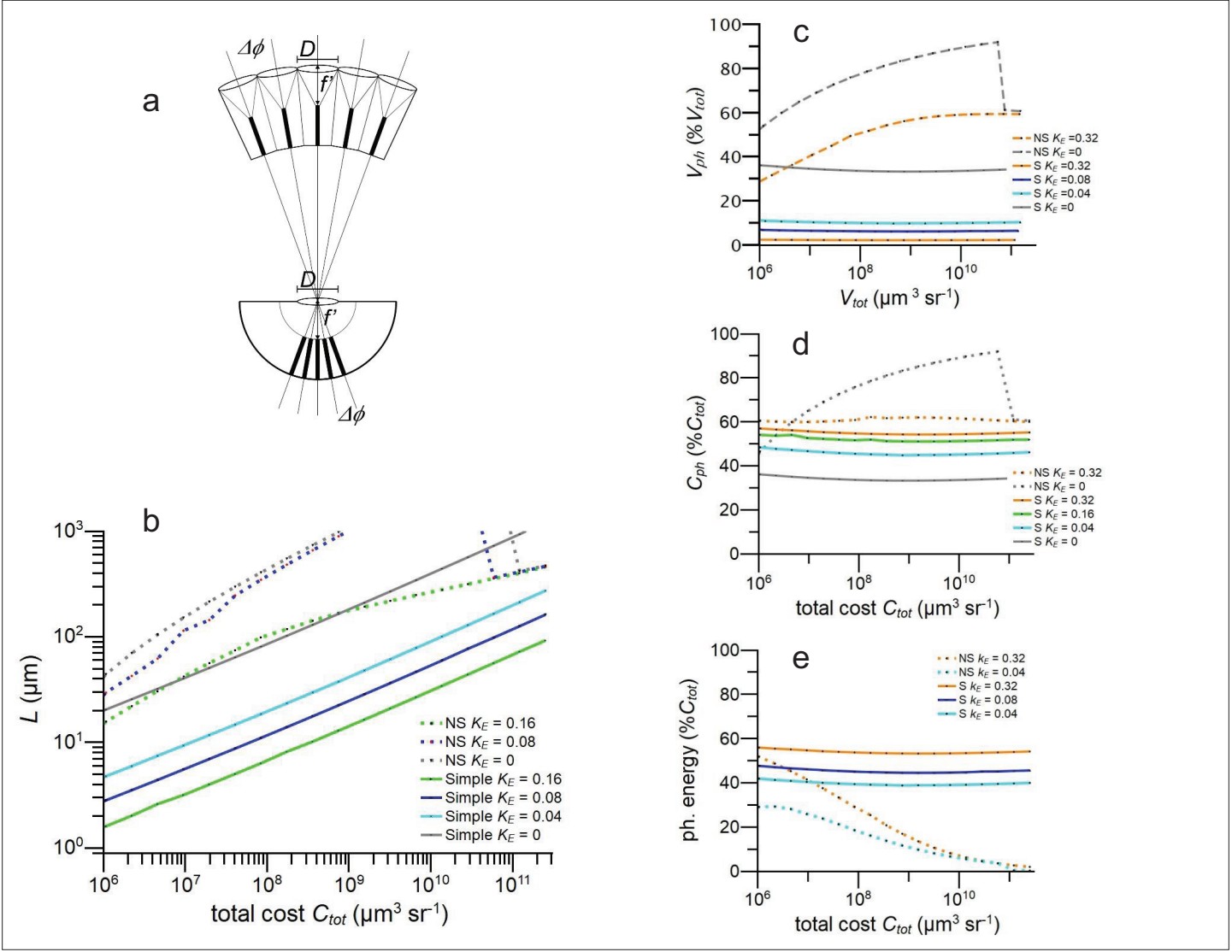

**Figure 7.** Resource allocation in model simple eyes optimised for information capacity compared to optimised model neural superposition apposition eyes (NS). (**a**) Schematic showing apposition eye and simple eye with identical spatial resolution, as defined by lens diameter $D$, focal distance $f'$, rhabdom(ere) diameter $d_{rh}$, and rhabdom(ere) length, $L$ (after **Kirschfeld, 1976**). Note the denser packing of photoreceptors inside the simple eye. (**b–e**) Properties of simple and NS eye models optimised for information capacity in full daylight. (**b**) Photoreceptor length, $L$ vs specific volume $V_{tot}$; (**c**) photoreceptor specific volume $V_{ph}$ expressed as $\%V_{tot}$ vs $V_{tot}$. (**d**) Photoreceptor investment $C_{ph}$ as $\%C_{tot}$ vs $C_{tot}$. (**e**) photoreceptor energy cost as $\%C_{tot}$ vs $C_{tot}$. Models use **Snyder, 1979** CoG approximation for acceptance angle. Data for all graphs is provided in **Figure 7—source data 1**.

The online version of this article includes the following source data for figure 7:

**Source data 1.** Related to **Figure 7b–e**.

## Information capacity and investment

In all optimised eye models, apposition and simple, the spatio-temporal information capacity, $H_{opt}$, increases sub-linearly with total investment, $C_{tot}$ (**Figure 8a**), making a bit of information more expensive in a larger eye. In simple eyes, $H_{opt}$ increases as $\left(C_{tot}\right)^{0.8}$ across the full range of $C_{tot}$ while in apposition models the exponent is lower, decreasing from 0.6 to 0.5 as $C_{tot}$ increases. Thus, the simple eye model is 10 times more efficient when $C_{tot} = 1 \times 10^6$ µm³ sr⁻¹ and approximately 100 times more efficient when $C_{tot} = 1 \times 10^{11}$ µm³ sr⁻¹ (**Figure 8b**). Increasing the photoreceptor energy tariff reduces the efficiency of all eye types, but this effect declines as apposition eye models increase in size (**Figure 8b**) because, for reasons of geometry, volume costs marginalise photoreceptor energy cost.

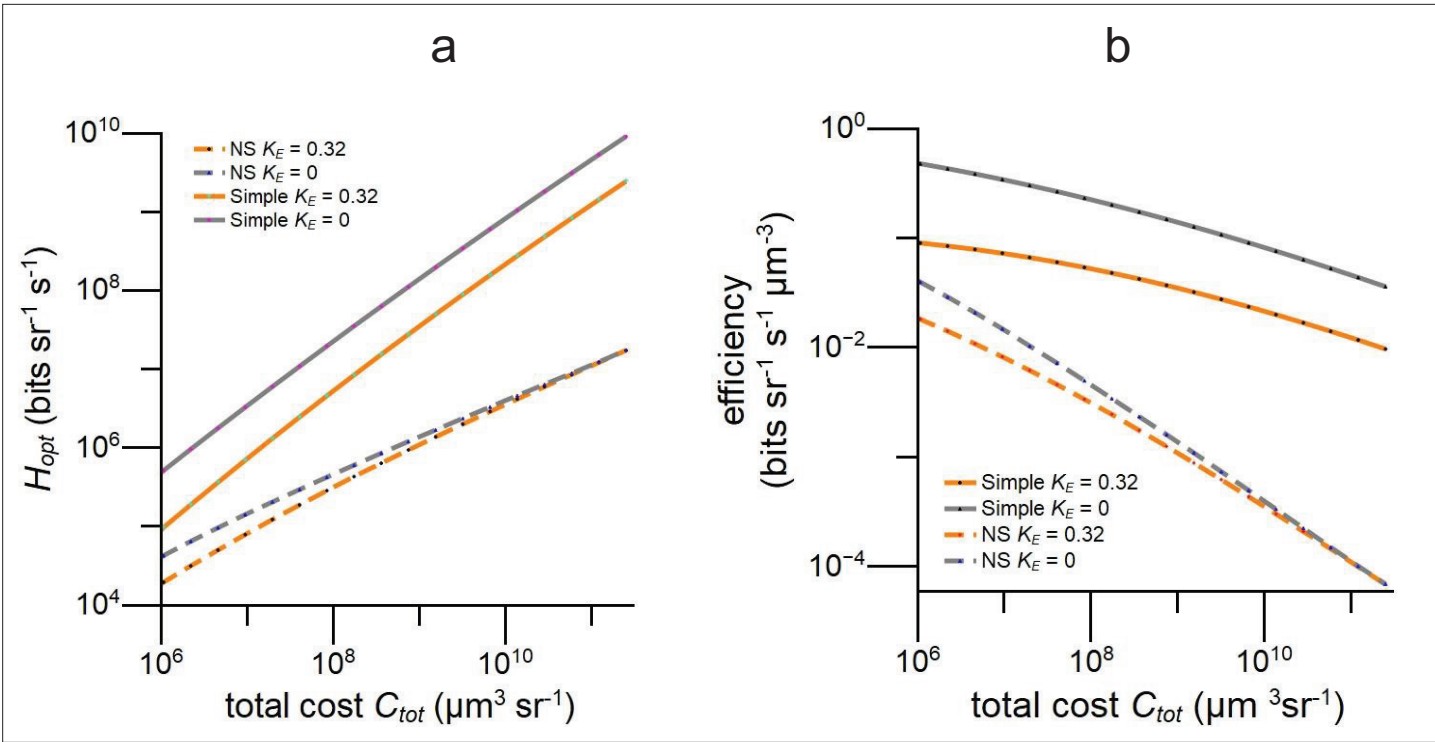

**Figure 8.** Information rate, $H$ vs total cost $C_{tot}$ for simple and apposition model eyes optimised to maximise information capacity in full daylight. (**a**) Rates are higher in simple eyes and more sensitive to photoreceptor energy tariff, $K_E$. (**b**) Efficiency, bits per unit specific volume per second, falls less steeply with increasing $C_{tot}$ in simple eyes. Models use Snyder's CoG approximation (1979) for acceptance angle. Data for all graphs is provided in *Figure 8—source data 1*.

The online version of this article includes the following source data for figure 8:

**Source data 1.** Related to *Figure 8a, b*.

## Discussion

We initiate a new approach to understanding eye design by investigating how the division of resources between an eye's dioptric apparatus and photoreceptor array influences an eye's performance and efficiency. To achieve this, we formulate a new measure of cost, specific volume, that is defined by an eye's shape and the dimensions of its components, can be adjusted to account for the high energy cost of photoreceptors, and relates investments in optics and investments in photoreceptors to three basic determinants of image quality, interreceptor angle $\Delta\phi$, photoreceptor acceptance angle $\Delta\rho$, and photoreceptor signal to noise ratio $SNR_{ph}$, via optical, geometrical, and physiological constraints. These advantages allow us to model how an eye's performance changes when resources are transferred between optics and photoreceptors. Thus, we can, for the first time, model performance across the morphospace of an eye of given type and total cost. Using a general-purpose measure of performance, spatio-temporal information capacity (*Howard and Snyder, 1983*; *van Hateren, 1992*), we find that the surface for an apposition compound eye (*Figure 3*) is flat topped. Thus, the configuration that maximises performance sits in a broad high-efficiency zone in which performance is more than 95% of maximum. Within this robust region of morphospace, an eye, or eye region, can be specialised for a specific task with little loss of information, and this can facilitate evolution by reducing loss of function. Our models predict that efficiently configured diurnal apposition eyes invest heavily in deep photoreceptor arrays with long rhabdomeres and rhabdoms, and match investments in optics and photoreceptor arrays so that rhabdom(ere) length increases with spatial resolution (*Figure 4*). By applying our new method of costing investments to published data, we show that fast-flying diurnal insects conform to these trends and are, therefore, configured to be efficient (*Figures 5 and 6*). Our models also demonstrate that photoreceptor energy consumption is an important factor in an apposition eye's design and performance: it reduces photoreceptor length by as much as 90% and information capacity by as much as 60%. To see how resource allocation and performance depend on

eye type, we model a simple (camera type) eye (*Figure 7*). When optimised for efficiency, a simple eye has much shorter photoreceptors and its structure and function are more sensitive to photoreceptor energy demands: photoreceptor length is greatly reduced by and information capacity decreases by as much as 90% (*Figures 7 and 8*). We conclude that when apposition and simple eyes are designed to use resources efficiently, the cost of the photoreceptor array is as significant as the cost of optics, and matching investments in optics and photoreceptor arrays to increase efficiency is a principle of eye design. Our analysis, which is the first to account for the costs of both optics and photoreceptor array, indicates that one benefit of investing in an eye, the ability to gather more information, follows the law of diminishing returns. As expected from its superior geometry (*Kirschfeld, 1976*), a simple eye is in theory a better investment: its information capacity increases as (total cost)$^{0.8}$, compared with (total cost)$^{0.55}$ for an apposition eye, and the simple eye is 10–100 times more efficient at gathering information (*Figure 8*). We now discuss the validity of our methodology, the robustness of our findings, and their contributions to our understanding of eye design and evolution.

## Specific volume is a useful measure of cost

This new measure of cost has several advantages. First, specific volume is defined by dimensions and angles that are measured anatomically and optically (*Equations 1, 2, 5, and 6*) and relates directly to benefits via optical, geometrical and physiological constraints on spatial acuity and sensitivity. Thus, like previous indicators of cost, namely eye radius, eye length, pupil area, and corneal area (*Brandon et al., 2015*; *Brandon and Dudycha, 2014*; *Snyder et al., 1977*; *Thomas et al., 2020*; *Thomas et al., 2004*), specific volume can evaluate investments and trade-offs that adapt eyes for specific purposes. Second, specific volume is more flexible and far-reaching than previous indicators of cost because it can be applied to eyes that are not well approximated by solid spheres. Third, it applies to small eye regions. Fourth, it provides a common currency for comparing investments in different eyes, eye regions, and components within an eye (*Figures 5–7*). This facility enables one to investigate how and why resources are distributed across visual fields and, as we demonstrate here, evaluate trade-offs between investments in an eye's components. Given that biological processes and systems evolve to become more cost-efficient because costs depress fitness (*Alexander, 1996*; *Niven and Laughlin, 2008*; *Taylor and Thomas, 2014*), we suggest a fifth advantage: specific volume will help us understand how eyes are adapted and evolve.

How well does specific volume represent costs? Volume accurately measures one limiting resource, space, but because there are insufficient data, we have to make assumptions to calculate the costs of energy and materials. Assuming that the densities of an eye's cells, extracellular materials and fluid-filled cavities are close to 1, space is a reasonable proxy for three costs that increase in proportion to mass: the energy expended carrying an eye (carriage cost), materials, and cellular energy consumption (*Witter and Cuthill, 1993*).

Equating mass and carriage cost is a reasonable starting point. Experiments on individual animals – a human walker carrying a backpack (*Bastien et al., 2005*), a flightless ant carrying nectar (*Duncan and Lighton, 1994*), and a flying beetle carrying a heavy weapon (*Goyens et al., 2015*) – show that the cost of carrying an additional load is the load's mass times the individual's mass specific metabolic rate when moving unloaded.

Turning to the costs of the materials and energy used within the eye, there is no data for an apposition eye's dioptric apparatus. Nonetheless, its corneal lenses, pigment cells, and cone cells are densely packed with macromolecules, and because their optical function is passive (they refract and absorb), they contain very few, if any, mitochondria. Indeed, the corneal cuticle and the fluid-filled pseudo-cone of fly NS eye are extracellular (*Hardie, 1985*). Thus, the assumption that the costs of materials and energy for the dioptric apparatus increase in fixed proportion to volume is a reasonable starting point.

Cost estimates can be adjusted for situations in which costs per unit volume are not equal, as illustrated by our treatment of photoreceptor energy consumption. To support transduction, the photoreceptor array has an exceptionally high metabolic rate (*Laughlin et al., 1998*; *Pangršič et al., 2005*; *Niven et al., 2007*). We account for this higher energy cost by using the animal's specific metabolic rate (power per unit mass and hence power per unit volume) to convert an array's power consumption into an equivalent volume (Methods). Photoreceptor ion pumps are the major consumers of energy and the smaller contribution of pigmented glia (*Coles, 1989*) is included in our calculation of the

energy tariff $K_E$ (Methods). The higher costs of materials and their turnover in the photoreceptor array can be added to the energy tariff $K_E$ but given the magnitude of the light-gated current (*Laughlin et al., 1998*), the relative increase will be very small. Thus, for our intents and purposes, the effects of these additional costs are covered by our models. For want of sufficient data, $K_E$ is uncertain; therefore, we model a reasonable range of values (Methods). Because the optimum eye configuration is most sensitive to $K_E$ in the lower half of this range (*Figures 4 and 7*), we are unlikely to seriously underestimate the effects of photoreceptor costs on eye morphology and efficiency.

## Matching photoreceptor length to spatial acuity

The systematic increase in rhabdomere or rhabdom length, $L$, with spatial acuity observed in apposition eyes (*Figures 1, 5 and 6*) and simple eyes (see below), and predicted by our models (*Figures 4 and 7*), is an obvious manifestation of efficient resource allocation. Two studies of apposition eyes, one of dragonfly and the other of praying mantis (*Labhart and Nilsson, 1995*; *Rossel, 1979*), map the covariation of lens diameter, $D$, interommatidial angle, $\Delta\phi$, and rhabdom length, $L$, across an entire eye. Both show that when acuity is increased by enlarging the dioptric apparatus and increasing ommatidial density, that is by reducing $\Delta\phi$, $L$ is increased to improve $SNR_{ph}$. In both insects $L$ peaks in a fovea that is used to detect and capture prey, and in dragonfly dorsal fovea the 1.1 mm rhabdomeres (the longest light sensitive waveguides reported in any eye, diurnal or nocturnal) are adapted to improve the detection and localisation of small flying prey (*Labhart and Nilsson, 1995*; *Rigosi et al., 2017*). However, specialisation for prey capture is unlikely to explain why $L$ varies systematically with acuity across an entire eye.

We find that the trend to increase rhabdomere length with acuity is more generally advantageous. The trend is widespread: looking across 28 eye regions drawn from 10 species of fast-flying insect, $L$ increases with acuity over a 25-fold range of interommatidial angle (*Figures 5 and 6*). Modelling shows that matching $L$ to acuity improves a general measure of performance, spatial information capacity and the widespread covariation of $L$ and acuity observed in apposition eyes follows the trend predicted by modelling (*Figures 5 and 6*). Finally, because information capacity captures the interplay between three basic determinants of image quality, $\Delta\rho$, $\Delta\phi$, and $SNR_{ph}$, several tasks could benefit from an increase in $L$ with acuity.

Other factors can link $L$ to acuity. One possibility, developmental constraints impose inflexible scaling relationships on apposition eyes, can be dismissed. Scaling changes across apposition compound eyes (*Perl and Niven, 2016*; *Scales and Butler, 2016*; *Taylor et al., 2019*), and our empirical log–log plots of $L$ vs eye volume and $L$ vs $D$ continuously change slope (*Figure 5d, e*), as required for the efficient allocation of resources (*Figure 4c*). Head size is an obvious constraint: smaller insects have smaller heads; consequently, they will tend to have smaller eyes with lower acuity and shorter rhabdom(ere)s. The strongest evidence that lack of headroom reduces the length of photoreceptors comes from the fovea of a small insect, *Holcocephala* (*Wardill et al., 2017*), that is adapted for exceptionally high acuity (*Table 2*; *Figure 5d*). One must also consider the costs associated with enlarging the head by adding more eye, such as mechanical support, wind resistance, and conspicuousness. We suggest that more comparative studies are required to establish the influence of head size on eye design.

Then there is the difference in $L$ that is so obvious and widespread that it has been accepted without question – $L$ is generally much longer in diurnal apposition eyes than in diurnal simple eyes. Comparing eyes with rhabdomeric photoreceptors, $L$ ranges from 60 μm to 1100 μm in apposition eyes (*Tables 2 and 3*), and from 10 μm to 90 μm, in spiders' simple eyes (*Land, 1985*). Efficient resource allocation provides an explanation: optimised simple eye models predict that $L$ is approximately x10 shorter than in apposition compound eyes of the same cost (*Figure 7b*). $L$ also increases with acuity (increasing $D$ and decreasing $\Delta\phi$) in optimised simple eye models and there is evidence that this matching occurs in diurnal spiders. $L$ increases with increasing $D$ when comparing eyes in different species (*Land, 1985*), different eyes in the same individual, and the same eye increasing in size and acuity as a juvenile spider grows to adulthood (*Goté et al., 2019*). However, more accurate models of these spiders' eyes are needed to prove that this covariation is increasing efficiency. The degree to which the lengthening of cone outer segments in primate fovea (*Provis et al., 2013*) increases the efficiency of resource utilisation is also an open question. A critical ingredient, a physiological constraint that links investments in outer segments to noise levels in full daylight, has yet to be identified in ciliary

photoreceptors. In addition, the economics of vertebrate simple eyes differs from invertebrate simple eyes in two ways. First, a vertebrate outer segment is almost 10 times more effective at absorbing photons because its folded sheets of membrane pack visual pigment molecules more densely than arrays of tubes (microvilli) (*Fain et al., 2010*). Second, photoreceptors compete with neural circuits for resources within the eye (*Kröger and Biehlmaier, 2009*; *Sterling and Laughlin, 2015*).

## Photoreceptor costs are a major factor in eye design

In apposition eye models optimised for spatio-temporal information capacity, more than 40% of total investment, $C_{tot}$, is allocated to photoreceptor arrays (*Figures 4e and 6e*) and, in line with this theoretical finding, our analysis of data from diurnal insects shows that arrays occupy 30–75% of eye volume (*Figures 5f and 6d*). For simple eye models, the efficient allocation is 30–60% of $C_{tot}$ (*Figure 7d*). Our modelling identifies roles for two types of array cost, the rarely considered costs associated with array volume (*Baden and Nilsson, 2022*; *Kirschfeld, 1976*; *Kröger and Biehlmaier, 2009*), and the well-known energy costs associated with phototransduction (*Ames, 2000*; *Fain and Sampath, 2021*; *Laughlin et al., 1998*; *Okawa et al., 2008*; *Pangršič et al., 2005*). We find that this energy cost greatly influences the configuration of an efficient eye: increasing the energy tariff $K_E$ reduces array depth, $L$, as much as tenfold (*Figures 4c, 5d, 6c, 7b*). Reducing $L$ reduces $SNR_{ph}$ (*Equation 4*), however, the $SNR_{ph}$ that remains is worth paying for because the %$C_{tot}$ allocated to photoreceptor energy consumption increases (*Figures 4f and 7e*). We conclude that because $SNR_{ph}$ is valuable, photoreceptor costs are always significant and this necessitates efficient resource allocation. It follows that photoreceptor costs influence the morphologies of both photoreceptor array and dioptric apparatus because optics and array compete for the resources invested in an eye.

## The impact of photoreceptor costs depends on eye type

Photoreceptor energy cost has more impact on both the total cost and the design of a simple eye than an equivalent apposition compound eye because the simple eye has more photoreceptors per unit volume (*Figure 7a*). Applying our lowest energy tariff, $K_E = 0.04$ μm³ per microvillus, to an optimised apposition eye model has almost no effect on the depth of the photoreceptor array, $L$, but in the optimised simple eye model, it reduces $L$ by more than 80%, and array volume by more than 60% (*Figure 7c*). Despite this shrinkage, investment in the simple eye model's photoreceptor array increases from 35% $C_{tot}$ to 50% $C_{tot}$ because energy costs have risen from 30% $C_{tot}$ to 55% $C_{tot}$ (*Figure 7d*). Also, in simple eye models, the percentages of $C_{tot}$ allocated to the photoreceptor array volume and energy change very little as $C_{tot}$ increases, whereas in apposition eyes the relative contribution of photoreceptor energy consumption falls steeply as photoreceptors become increasingly widely spaced (*Figure 7a*). Thus, the distribution of costs within an efficiently configured eye changes with eye type, according to the eye's typical geometry.

## Efficiency depends on eye type and photoreceptor energy cost

In both apposition and simple eye models, efficiency decreases with increasing $C_{tot}$ (*Figure 8b*), making a bit of information more expensive in a larger eye. The decline in efficiency with increasing eye size is steeper in apposition models, as expected from the well-known scaling relationships enforced by lens diffraction and eye geometry: spatial resolution increases as eye radius, $R$, in a simple eye and as $\sqrt{R}$ in an apposition eye (*Kirschfeld, 1976*). However, using $R$ as a measure of investment fails to identify the contribution of the photoreceptor array and fails to account for the fact that an apposition eye is not a solid sphere. When our more realistic measure of cost, specific volume, is used, a *Drosophila* size apposition eye is 10 times less efficient than a simple eye of the same cost and a dragonfly size eye is 100 times less efficient (*Figure 8b*). In both simple and apposition eyes, efficiency is reduced by increasing the photoreceptor energy tariff $K_E$. This effect is much greater in simple eyes (*Figure 8b*); thus, as also found for the reduction of photoreceptor length (*Figure 7b*), $K_E$ has more impact on the design of simple eyes.

## Expanding our understanding of apposition eye design

We build on theoretical studies that show how constraints imposed by apposition optics, eye geometry, and eye size determine spatial resolution and information capacity when eye radius $R$, and hence corneal area, represent the limiting resource (*Howard and Snyder, 1983*; *Kirschfeld, 1976*; *Snyder*

*et al., 1977*). *Howard and Snyder, 1983* introduced the physiological constraint, transduction unit saturation, and showed that it increases the optimum value of eye parameter, $p$, in bright light by placing a fixed ceiling on $SNR_{ph}$ that is equivalent to operating at a lower light level. We extended their approach by using a measure of cost, specific volume, that allows eye radius $R$ to vary while total cost remains constant. This added realism allows us to raise and lower the $SNR$ ceiling imposed by transduction units by trading investments in rhabdomere length against investments in dioptric apparatus. In so doing, we discover that it is advantageous for photoreceptors to lengthen rhabdomeres so as to reduce shot noise. Indeed, in daylight, *Calliphora vicina* R1–6 achieve the lowest level of photoreceptor noise reported to date, an equivalent contrast of 0.0084 (*Anderson and Laughlin, 2000*). This extreme specialisation resembles Autrum's proposition that apposition eye photoreceptors compensate for poor spatial resolution by increasing temporal resolution (*Autrum, 1958*), thereby achieving the highest flicker fusion frequencies known, in excess of 400 Hz (*Tatler et al., 2000*). Perhaps enhancing photoreceptor performance to compensate for optical inefficiency is a principle of apposition eye design?

Increasing temporal resolution increases photoreceptor energy consumption, so to improve efficiency, apposition eye photoreceptors match temporal bandwidth to the properties of the images they form (*Niven et al., 2007*; *Laughlin and Weckström, 1993*). Eye regions that have lower spatial resolution and encounter more slowly moving images save energy by reducing photoreceptor bandwidth (*Burton et al., 2001*). Although this matching makes large energy savings, it is not included in our models because, to reduce the burden of establishing proof of principle, we fix photoreceptor bandwidth at the high value measured in a fully light-adapted blowfly. Nonetheless, our models can accommodate the costs and benefits of changing temporal bandwidth because they operate in the frequency domain (Methods). Boosting temporal resolution will be more cost-effective in large apposition eyes because photoreceptor energy costs are marginalised by costs associated with array volume (*Figure 4e, f*), therefore investments of energy that increase temporal bandwidth are increasing the return on the much larger investment in array volume (e.g. carriage cost). Some fast-flying diurnal insects with large facet lenses reduce the carriage cost of the photoreceptor array by surrounding each ommatidium with a palisade of air-filled tracheae (*Horridge, 1969*; *Laughlin and McGinness, 1978*; *Smith and Butler, 1991*; *Wardill et al., 2017*). The volumes and savings involved can be large. Published micrographs of dragonfly dorsal eye (*Horridge, 1969*) suggest that air fills at least 50% of array cross section.

## How efficiently do insects allocate resources within their apposition eyes?

In principle, one can establish the efficiency of an insect's investments in optics and photoreceptors by computing the performance surface for a model eye region that has the same parameters as the insect's eye region, then observing how closely the insect's eye region approaches the optimum configuration. We cannot do this at present because a critical parameter, the photoreceptor energy tariff $K_E$, has not been determined: the best we can do is use data from blowflies to estimate $K_E$'s likely range (Methods). Not knowing $K_E$ is a major impediment to obtaining exact fits of our models to data because our modelling has discovered that the morphology of an efficient eye is sensitive to photoreceptor energy costs: in apposition eyes, increasing $K_E$ across its likely range reduces the depth of the photoreceptor array by as much as 90% (*Figure 4c*). Nor can we use our models to infer $K_E$ by adjusting it to fit the data. Changing $K_E$ has very little effect on $D$ and $\Delta\phi$, therefore we must fit the model to $L$, and this requires several data points from eyes of the same species because $K_E$ will be species specific (Methods). In our most consistent data set, the NS eyes (*Figure 5*; *Table 2*), it is impossible to obtain a satisfactory fit: the 12 data points come from 7 species and there are never more than 2 points per species. There are several data points per species for fused rhabdom apposition eyes (*Figure 6*; *Table 3*), but in each of the four species, the data points are too widely scattered to obtain a useful fit. This scatter in values of $L$ is not surprising. Our generic fused rhabdom model glosses over the large differences in F-number and rhabdom diameter seen within the eye of each species (*Table 3*), and these differences probably reflect the specialisation of eye regions for different purposes, for example in the fovea, the detection, tracking, and interception of mates and prey, and in the periphery, the monitoring of optic flow and the detection of potential prey and predators (*Menzel et al., 1991*; *van Hateren et al., 1989*; *Labhart and Nilsson, 1995*; *Straw et al., 2006*). This

proposition is supported by the observation that the empirical data for $D$ and $\Delta\phi$ (**Figure 6a, b**) are also more widely scattered than the data for the more homogeneous group, fly NS eyes (**Figure 5a, b**). Because our method calculates three basic determinants of the quality of the captured image, $\Delta\phi$, $\Delta\rho$, and $SNR_{ph}$, it can be adapted to compute performance surfaces for these specialised tasks; however, the necessary theory has not been developed.

Having raised the question, efficient for what, we must examine our assumption that every eye region of a diurnal insect is adapted for vision in full daylight. The fact that insects in our data sets have larger lenses, interommatidial angles and eye parameters than our models predict (**Figure 5**) suggests that their eye regions are adapted for lower light levels (**Snyder, 1979**). We can think of two mutually inclusive reasons why a diurnal eye will not be adapted for vision in full daylight. The eye might be adapted for the time of day when the behaviour it supports is most profitable, for example insects that are most active early and/or late in the day to avoid predators and desiccation, and insects that prey on these crepuscular insects. Second, because an eye's performance surface is flat-topped, an eye region optimised for an intermediate light intensity might, when averaged over the daylight hours, perform better than an eye adapted for bright light. Our models can be developed to investigate how costs and benefits influence the design of eyes adapted for lower light levels (see below), and modified to investigate two further reasons for increasing the eye parameter $p$, higher image speeds (**Snyder, 1979**) and increasing $D$ in 'bright zones' to improve the detection of small targets and small movements (**van Hateren et al., 1989**; **Straw et al., 2006**).

In summary, for want of both data and theory, we are currently unable to quantify the efficiency with which an apposition eye region performs the function for which it is adapted. Nonetheless, we have two reasons to conclude that flying diurnal insects allocate resources to the optics and photoreceptor arrays of their apposition eyes efficiently. First, the empirical values we extract from published data lie within or close to the boundaries defined by our theoretical curves for eyes that maximise a general-purpose measure of performance, information capacity (**Figures 5 and 6**). Second, the flat-topped performance surfaces have extensive high-efficiency zones within which an eye's morphology can be changed while retaining >95% of the maximum information capacity (**Figure 3**). Thus, eyes that are not optimised to maximise information capacity can still be gathering information with high efficiency. This observation speaks to the value of deriving performance surfaces: one can consider competing demands, evaluate loss of function, and identify compromises.

## Efficient resource allocation is a general principle

Efficient resource allocation is observed at many levels of biological organisation, from molecules to life histories (**Bigman and Levy, 2020**; **Garland et al., 2022**; **Nijhout and Emlen, 1998**; **Stearns, 1989**). Eyes are no exception. In an apposition eye, optical radius, $R$, represents corneal surface area per unit solid angle, therefore models that maximise spatial acuity or information capacity at constant $R$ show that corneal area can be allocated to facet lenses to maximise spatial acuity and information capacity by trading spatial resolution for $SNR$ (**Snyder, 1977**; **Snyder et al., 1977**). In the POL region of fly NS eye, a fixed resource, the number of microvilli in the central rhabdom, is allocated to a pair of photoreceptors, R7 and R8, to increase the efficiency of polarisation coding (**Heras and Laughlin, 2017**), and in primate eye, retinal area is allocated to cone spectral types to increase coding efficiency (**Garrigan et al., 2010**; **Zhang et al., 2022**).

Matching the sampling density of the photoreceptor array to lens resolving power (**Nilsson and Land, 2012**) also allocates the resources invested in photoreceptors more efficiently by avoiding wasteful overcapacity, according to the principle of symmorphosis (**Piersma and Gils, 2011**; **Taylor and Weibel, 1981**; **Weibel, 2000**). However, unlike the matching of $L$ to $\Delta\phi$ that allocates resources efficiently to optics and photoreceptors, the matching of capacities does not consider the individual cost–benefit functions of a system's components. When these are taken into account, the allocation of investment that maximises efficiency can instal overcapacity in components with more favourable cost–benefit ratios, as demonstrated for bones in racehorse forelegs (**Alexander, 1997**) and chains of signalling molecules (**Schreiber et al., 2002**).

## Understanding the adaptive radiation of eyes

The method we introduce here can be developed to view several aspects of eye adaptation and evolution through a cost–benefit lens because it is flexible. Our approach is not wedded to information

capacity: it supports other measures of visual performance because it relates investments to fundamental determinants of image quality. Moreover, the method can be applied to several trade-offs that influence eye design by relaxing the simplifying assumptions we make here. To give a concrete example, the trade-offs between acuity and sensitivity that adapt eyes for nocturnal or diurnal vision can be viewed through a cost–benefit lens by allowing $F$-number and rhabdom(ere) diameter to vary, and by adopting a method that calculates the mean and variance of photon transduction rate on $N_{vil}$ in rhabdomeres of any given length and variable cross section, at any biologically plausible light intensity (*Heras and Laughlin, 2017*).

One can also relax the restrictions we place on eye geometry. Our approach does not depend on spherical geometry; it depends on being able to relate the shapes, sizes and constituents of an eye's components to their costs and performance. Advances in scanning, imaging and digital reconstruction offer opportunities to determine shapes, volumes and constituents directly, and advances in modelling optical and physiological processes are extending our ability to relate these properties to performance (*Kim, 2014*; *Song et al., 2012*; *Sumner-Rooney et al., 2019*; *Taylor et al., 2019*). These powerful new methods can take us closer to the goal of understanding how costs, benefits and innovative mechanisms drive eye evolution (*Sumner-Rooney, 2018*), and our flexible cost–benefit approach can support this endeavour.

Our approach also identifies factors that contribute to selective pressure and gauges their effects. For example, the pressure to reduce photoreceptor energy cost depends on a photoreceptor's impact on an animal's energy budget. This effect is captured by our photoreceptor energy tariff, $K_E$, because it relates a photoreceptor's energy consumption to the animal's metabolic rate (Methods). We also find that photoreceptor energy costs are marginalised when apposition eye size increases, and this encourages the use of wide bandwidth, high $SNR$ photoreceptors with high metabolic rates. Most interestingly, the derivation of performance surfaces (*Figure 3*) reveals a large area, the robust high-performance zone within which an eye's morphology can be adapted to a particular task while retaining more than 95% of a measure of general-purpose capability, information capacity. This low 'loss of opportunity' cost helps to explain the remarkable degree of fine tuning observed in apposition compound eyes (*Burton et al., 2001*; *Land, 1997*; *Nilsson and Land, 2012*; *Taylor et al., 2019*; *Sumner-Rooney et al., 2019*), in line with suggestions that robustness increases evolvability (*Drack and Betz, 2022*).

Two key steps in the early evolution of eyes were the stacking of photoreceptive membranes to absorb more photons, and the formation of optics to intercept more photons and concentrate them according to angle of incidence (*Nilsson, 2013*; *Nilsson, 2021*). Our modelling of well-developed image forming eyes shows that to improve performance, stacked membranes (rhabdomeres) compete with optics for the resources invested in an eye, and this competition profoundly influences both form and function. Thus, it is likely that competition between optics and photoreceptors was shaping eyes as lenses evolved to support low resolution spatial vision. If so, the developmental mechanisms that allocate resources within extant high-resolution eyes (*Casares and McGregor, 2021*), by controlling cell size and shape and, as our study emphasises, setting up gradients of shape and size across the eye, will have analogues or homologues in ancient eyes. Their discovery will not only tell us how Nature constructs perfectly contrived instruments, it will illuminate the evolutionary pathway to perfection.

## Methods
### Calculating a photoreceptor array's information rate
Our calculations of the rate at which an eye's photoreceptor array codes achromatic information, in bits per steradian per second, take account of five factors. (1) The statistics of the signals presented to an eye under natural conditions; (2) the blurring of the spatial image by the eye's dioptric apparatus; (3) the sampling of the spatial image by the photoreceptor array; (4) the introduction of noise by the stochastic activation of a finite population of transduction units; and (5) the temporal smoothing of signals and noise during phototransduction. We calculate information rates in the frequency domain using methods developed for apposition eyes (*de Ruyter van Steveninck and Laughlin, 1996*; *van Hateren, 1992*; *Howard and Snyder, 1983*; *Snyder et al., 1977*).

## Natural intensity signals

We calculate the spatial and temporal power spectra of naturalistic signals using the model established for blowfly (*van Hateren, 1992*). An eye images an ensemble of natural scenes in which power decreases as the square of spatial frequency, as measured and predicted theoretically (*Burton and Moorhead, 1987*; *van Hateren, 1992*; *Ruderman, 1997*). Consequently, the two-dimensional power spectrum of the spatial frequencies viewed by the eye is

$$S_{xy} = \frac{c_c}{f_x^2 + f_y^2}.$$  (12)

The proportionality constant $c_c$ is chosen to set the total contrast of the image formed by the eye (defined as the square root of the ratio between the power of the signal and the square of the mean level of signal) to an appropriate value. This contrast depends on the eye that observes the scene. For images spatially filtered by a blowfly compound eye, the average contrast, calculated locally over intervals of 25–50°, is 0.4 (*Laughlin, 1981*). Accordingly, we use a proportionality constant that scales the power of frequencies between one cycle per 50° and the upper cut-off frequency dictated by sampling, one cycle per 1.5°, to be $(0.4)^2 = 0.16$.

To add the temporal dimension *van Hateren, 1992* modelled how the movement of an eye through an ensemble of uniformly distributed objects generates temporal structure. For the case of an eye moving in a straight line, the structure is adequately described by a random distribution of angular speeds

$$a_v(v) = \frac{c_v}{(|v| + \sigma_v)^2},$$  (13)

where $a_v(v)$ is the probability of encountering image speed $v$, $c_v$ is the normalisation constant that sets the sum of probabilities over the distribution to 1, and $\sigma_v$ is a constant that regulates the width of the distribution. Local movements produced by objects moving within a scene, although acknowledged to be biologically significant, have little impact on the statistics of the signal encountered by an eye moving in a straight line (*van Hateren, 1992*).

Distributions of this form peak at lower speeds and have a long tail that extends to speeds faster than $\sigma_v$. The exact value of $\sigma_v$ will depend on the behaviour of the animal. For *C. vicina*, yaw, pitch and roll movements rarely exceed $3.5\,\mathrm{rad\,s^{-1}}$ between saccades (*van Hateren and Schilstra, 1999*). Assuming a similar limit in translational movements, we take $\sigma_v$ to be $1\,\mathrm{rad\,s^{-1}}$. In his earlier 1992 paper, van Hateren assumed $\sigma_v$ is $0.29\,\mathrm{rad\,s^{-1}}$ and noted that the information capacities he calculated are not particularly sensitive to $\sigma_v$.

The three-dimensional power density of naturalistic signals generated by the aforementioned distributions is

$$S_{xyt}(f_x, f_y, f_t) = \frac{\pi}{2f_r^2} c_c a_v \left( \frac{\pi}{2} \frac{f_t}{f_r} \right),$$  (14)

where $f_r = \sqrt{f_x^2 + f_y^2}$ and $c_c$ are the proportionality constants used in *Equation 12*. We checked that this distribution retains the local contrast value of 0.4 for different values of the parameter $|\sigma_v|$.

## Blurring by the optics

The lens point-spread function and the angular sensitivity of the photoreceptor entrance aperture combine to determine photoreceptor angular sensitivity, which in the eyes we consider approximates a Gaussian function of half-width $\Delta\rho$. The transfer function of this spatial filter is

$$m(f_x, f_y) = \exp \left( -\frac{\pi^2}{4 \ln 2} (f_x^2 + f_y^2) \Delta\rho^2 \right).$$  (15)

The lens is diffraction limited and the photoreceptive rhabdomeres and rhabdoms are waveguides; consequently, calculating the acceptance angle of this coupled system is a complicated problem of wave optics, especially when, with narrow $d_{rh}$, waveguide effects are significant (*Stavenga, 2003a*;

*Stavenga, 2003b*, *Stavenga, 2004a*; *Stavenga, 2004b*). We employ two expressions for photoreceptor acceptance angle. One is Snyder's CoG approximation

$$\Delta\rho^2 = (\lambda/D)^2 + \Delta\rho_r^2,$$

(16)

where $\lambda$ is the wavelength of light, $D$ is lens diameter, $\lambda/D$ is the half-width of the Airey disk, and $\Delta\rho_r$ is the angular acceptance of the rhabdomere, taken to be its angular subtense, $\Delta\rho_r = d_{rh}/f$ (*Snyder, 1979*).

The other is Stavenga's more principled expression, obtained using his wave optics model (WOM) of larger fly NS eyes (*Stavenga, 2004a*)

$$\Delta\rho = 1.26\lambda/D.$$

(17)

Stavenga's WOM shows that the coefficient in *Equation 17* reduces progressively as the fly's longitudinal pupil closes, reaching a minimum of 1.14 when fully closed in full daylight. We use an intermediate value, 1.26, because the pupil closes progressively over the upper two log units of daylight intensity, while holding $SNR_{ph}$ close to maximum.

## Temporal filtering during phototransduction

A photoreceptor's elementary response to a single photon, a quantum bump, is generally fitted with functions of the type

$$h(t) \propto \left(\frac{t}{\tau}\right)^{\alpha-1} e^{-t/\tau},$$

(18)

whose Fourier transform is

$$H(f) = \frac{1}{(1 + i2\pi\tau f)^\alpha}.$$

(19)

Thus, transduction's first temporal low-pass filter has the form

$$|m_t(f_t)|^2 = \frac{1}{(1 + (2\pi\tau f)^2)^\alpha}.$$

(20)

We use the fit of $\alpha = 3.12$ measured in *Musca* (*Burton, 2006*) and $\tau = 0.001\,\text{s}$ obtained in *Calliphora* at the highest light intensities (*Juusola et al., 1994*). According to these values, photon shot noise power falls to half-maximum at around 170 Hz.

Following the activation of a microvillus by an absorbed photon, the quantum bump is produced with a variable delay. This latency dispersion low-pass filters signals that are composed of many bumps by widening the impulse response. We use the low-pass filter

$$|m_d(f_t)|^2 = \frac{1}{(1 + (2\pi\tau_d f)^2)^{\alpha_d}},$$

(21)

where $\alpha_d = 2$ (*Wong et al., 1980*). With $\tau_d = 0.0014\,\text{s}$ this filter produces a signal corner frequency close to the 55 Hz measured in *C. vicina* (*Anderson and Laughlin, 2000*). The trends we observe in our study are not critically dependent on the exact values of these bump parameters.

## Sampling

For one-dimensional receptor arrays, frequencies will not be correctly sampled above the Nyquist limit (*Snyder, 1979*)

$$f_{Ny} = \frac{1}{2\Delta\phi},$$

(22)

where $\phi$, as defined above, is the sampling angle of the array.

For a two-dimensional hexagonal lattice, only frequencies up to $\frac{1}{\sqrt{3}\Delta\phi}$ can be sampled independently of the orientation of the array (*Snyder et al., 1977*). These authors also noted that frequencies up

to $\frac{1}{\Delta\phi}$ can be sampled for special orientations of the array; however, we will assume that the highest spatial frequencies sampled by an array is $\frac{1}{\sqrt{3}\Delta\phi}$.

## Deriving the signal coded by the array

To obtain the power spectral density of the signal coded by a photoreceptor, the natural image signal derived above is spatially filtered by the dioptric apparatus, converted to a stream of shot events, quantum bumps, with mean rate $\psi$, and temporally filtered during phototransduction. We start with the **van Hateren, 1992** expression for the signal coded by a photoreceptor that is transducing photons at a rate $\psi$

$$\left(\Delta S\right)^2 (f_x, f_y, f_t) = \psi^2 S_{xyt}(f_x, f_y, f_t)|m(f_x, f_y)|^2|m_t(f_t)|^2|m_d(f_t)|^2. \tag{23}$$

This expression assumes that the photoreceptor signal increases linearly with photon capture rate, as happens at low light levels when the saturation of transduction units is negligible, and signal and noise follow Poisson statistics.

In brighter conditions, the saturation of microvilli changes the relationship between transduction rate, $\psi$ and photoreceptor signal $\Delta S$ (**Howard et al., 1987**). The probability that a microvillus generates a signal is given by

$$p = \frac{I}{N_{vil}/\tau_r + I}, \tag{24}$$

where $I$, the effective intensity, is the rate at which microvilli are absorbing photons, and the cycling time $\tau_r$ is the average time taken to reset a microvillus to its receptive state, following activation by a photon capture (**Howard et al., 1987**).

The transduction rate is given by

$$\psi = pN_{vil}/\tau_r. \tag{25}$$

There are some limits to this treatment. Because light intensity decreases along the rhabdom(ere), microvilli in the distal part will, in principle, absorb more photons than microvilli in the proximal part, making it impossible for all microvilli to have the same probability of being active. However, adaptations have been described that partially compensate for this effect by reducing the quantum capture efficiency of distal microvilli (**Labhart and Nilsson, 1995**).

According to the saturation **equation (24)**, the light intensity that activates half the microvilli, $I_{50} = N_{vil}/\tau_r$. Therefore, the dependence of transduction rate on $I$ is given by

$$\psi = pN_{vil}/\tau_r = \frac{I_{50}I}{I_{50} + I}, \tag{26}$$

and

$$\frac{d\psi}{dI} = \frac{I_{50}^2}{(I_{50} + I)^2}. \tag{27}$$

$\Delta\psi$, the change in $\psi$ produced by a change in intensity, $\Delta I$, is given by

$$\Delta\psi = \frac{d\psi}{dI}\Delta I = \frac{d\psi}{dI}cI, \tag{28}$$

where $c$ is the stimulus contrast (**Howard et al., 1987**)

Under the condition we model, half-saturation of transduction units, $I = 2\psi$ and $\frac{d\psi}{dI} = \frac{1}{4}$. Consequently, the photoreceptor signal is

$$\left(\Delta S\right)^2 (f_x, f_y, f_t) = \left(\frac{\psi}{2}\right)^2 S_{xyt}(f_x, f_y, f_t)|m(f_x, f_y)|^2|m_t(f_t)|^2|m_d(f_t)|^2, \tag{29}$$

where

$$\psi = \frac{N_{vil}}{2\tau_r}.\qquad(30)$$

In the fly NS eye, six photoreceptors sample the same point in space, therefore the signal transmitted for each pixel in the array is

$$\left(\Delta S\right)^2 (f_x, f_y, f_t) = \left(3\psi\right)^2 S_{xyt}(f_x, f_y, f_t) |m(f_x, f_y)|^2 |m_t(f_t)|^2 |m_d(f_t)|^2.\qquad(31)$$

## Noise

We consider shot noise generated at the first step in the generation of a quantum bump, the absorption of a photon by a rhodopsin molecule in a microvillus that is not currently processing a bump. Noise generated elsewhere in photoreceptors is not incorporated in our model. Assuming that the photon capture rate $\psi$ is never far from its average value, the shot noise power in a single photoreceptor is given by:

$$\sigma_{sn}^2(f_t) = \frac{\psi}{2} |H(f_t)|^2,\qquad(32)$$

where $|H(f_t)|$ is the Fourier transform of the quantum bump and the factor 2 takes account of the fact that at half-saturation the statistics of microvillus activation are binomial (**Howard et al., 1987**).

Substituting from **Equation 20**

$$\sigma_{sn}^2(f_t) = \frac{\psi}{2} |m_t(f_t)|^2.\qquad(33)$$

We assume that the shot noise in different photoreceptors is uncorrelated, therefore the shot noise generated by the six photoreceptors coding a pixel is

$$\sigma_{sn}^2(f_t) = 3\psi |m_t(f_t)|^2.\qquad(34)$$

Because we assume that the shot noise in different photoreceptors is uncorrelated, the power spectrum of shot noise is on average the same across the photoreceptor array

$$\sigma_{sn}^2(f_x, f_y, f_t) \propto \begin{cases} 3\psi |H(f_t)|^2 & \text{if } (f_x, f_y) \text{ is sampled} \\ 0 & \text{if } (f_x, f_y) \text{ is not sampled} \end{cases}\qquad(35)$$

with the proportionality constant being the inverse of the area in frequency space that is sampled by the photoreceptor array.

## Aliasing

When the image projected to the photoreceptor array contains spatial frequencies that are beyond the spatial resolving power of the array, these are aliased to lower frequencies, thereby distorting the sampled image (**Snyder et al., 1994**). We assume that all aliased spatial frequencies act as noise, shared equally between the correctly sampled spatial frequencies:

$$\sigma_{al}^2(f_t) = \psi^2 k(f_t).\qquad(36)$$

## Calculating information

We calculate the spatial information gathered by the photoreceptor array by modifying the formula for a square spatial sampling lattice (**Howard and Snyder, 1983**). For a hexagonal array, the equation for spatial information has the form

$$H = A \int_0^{\mu_s} f_r \log_2[1 + (S/N)(f_r)] df_r,\qquad(37)$$

where the array cut-off frequency is $\mu_s = \frac{1}{\sqrt{3}\Delta\phi}$. To obtain the factor $A$, we consider the extreme case where the signal-to-noise ratio is constant across all spatial frequencies, in which case

$$H = A \frac{1}{3(\Delta\phi)^2} \log_2[(N_i)], \tag{38}$$

where $N_i = \sqrt{1 + S/N}$ is the number of discriminable levels of contrast coded by each ommatidium. The factor multiplying $\log_2[N_i]$ is the number of ommatidia in a steradian, which for a hexagonal lattice is $1/S_{hexagon} = \frac{2\sqrt{3}}{3(\Delta\phi)^2}$. Thus, $A = 2\sqrt{3}$. Therefore spatial information capacity per steradian is

$$H = A \int_0^{\mu_s} f_r \log_2[1 + (S/N)(f_r)]df_r. \tag{39}$$

We then extend this equation for spatial information capacity to the temporal domain (*van Hateren, 1992*). For a spatio-temporal signal, the information capacity is

$$H = 2\sqrt{3} \int_{-\infty}^{\infty} \int_0^{\mu_s} f_r \log_2[1 + (S/N)(f_r, f_t)]df_t df_r. \tag{40}$$

Inserting signal and noise

$$H = 2\sqrt{3} \int_{-\infty}^{\infty} \int_0^{\mu_s} f_r \log_2(1 + \frac{(\Delta S)^2 (f_r, f_t)}{\sigma_{sn}^2(f_r, f_t) + \sigma_{al}^2(f_t)})df_t df_r. \tag{41}$$

## Ballpark estimate of the photoreceptor energy tariff

### Energy consumed per microvillus

Our estimates are based on high-resolution measurements of the oxygen consumed by isolated *C. vicina* compound eyes, made using a magnetic diver balance (*Pangršič et al., 2005*). In bright light, an eye consumes $1.8 \times 10^{-9}$ litres of oxygen per second which, converting to the units commonly used to express metabolic rates, is 6.48 mm³ of oxygen per hr. With 5000 ommatidia, 60,000 microvilli in the rhabdomere of a photoreceptor R1–6, and taking the central rhabdom formed by R7 and R7 to be equivalent to a single R1–6, the eye contains $2 \times 10^9$ microvilli, giving an oxygen consumption per microvillus in bright light $O_{vil}^L = 3.24 \times 10^{-9}$ mm³ oxygen per microvillus per hr. In darkness the eye's oxygen consumption drops to one third, giving $O_{vil}^D = 1.08 \times 10^{-9}$ mm³ oxygen per microvillus per hr.

Because these estimates are based on measurements from whole eyes, they include consumption by all cell types. There are three reasons why photoreceptors dominate total consumption. First, the values of an eye's oxygen consumption in light and in darkness are close to the total photoreceptor consumption predicted from electrophysiological estimates of the ATP consumed by single photoreceptors (*Laughlin et al., 1998*; *Pangršič et al., 2005*). Second, photoreceptors are the compound eye's most active cells because transducing photons at high light intensities necessarily requires large fluxes of ions and chemical intermediates. Third, to support their activity, photoreceptors contain most of a compound eye's mitochondria.

The rate of oxygen consumption averaged over the course of 24 hr $\bar{O}_{vil}$, depends on the hours of daylight, *DL*

$$\bar{O}_{vil} = \frac{DL * O_{vil}^L + (24 - DL) * O_{vil}^D}{24} \text{mm}^3 \text{oxygen per mg per hr}. \tag{42}$$

### Energy consumed by the fly

To convert the energy consumption of the eye per microvillus into an equivalent volume we should ideally use the animal's average specific metabolic rate, measured under natural conditions, but to the best of our knowledge this field metabolic rate has not been measured in blowflies. Therefore, we combine a value of the specific metabolic rate in continuous flight that is towards the high end of the range measured in *Calliphora*, $sFMR = 30$ mm³ oxygen per mg per hr (*Yurkiewicz and Smyth, 1966*), with the resting metabolic rate measured in the blowfly *Phormia*, $RMR = 2.6$ mm³ oxygen per mg per hr (*Keister and Buck, 1961*; *Niven and Scharlemann, 2005*), according to the number of hours per day spent flying, $T_F$, to estimate the daily average specific metabolic rate

$$\overline{SMR} = \frac{T_F * sFMR + (24 - T_F) * RMR}{24} \text{mm}^3 \text{oxygen per mg per hr}. \tag{43}$$

**Table 4.** Dependence of energy surcharge $K_E$ on time spent flying $T_F$ and hours of daylight $D_L$, calculated for blowfly.

| $T_F$ (hr) | $D_L$ (hr) | $K_E$ (µm³/microvillus) |
|---|---|---|
| 12 | 12 | 0.13 |
| 12 | 16 | 0.15 |
| 2 | 12 | 0.44 |
| 2 | 16 | 0.52 |

### Estimating the energy tariff, $K_E$

Assuming that the density of the eye is 1

$$K_E = \frac{\overline{O_{vil}}}{SMR} \, \text{mm}^3 \text{of eye per microvillus} \tag{44}$$

To obtain a plausible range of values we calculate $K_E$ in four conditions (**Table 4**).

Note that, as expected, our measure of the impact of photoreceptor energy costs, $K_E$, depends on both the energy consumption per microvillus and the animal's metabolic rate: $K_E$ is increased by increasing daily consumption by photoreceptors and decreased by increasing the animal's overall metabolic rate by increasing time flying, $T_F$.

## Preliminary model of a simple eye

Our generic simple eye model has the same $F$-number $F = 2$ as our generic fly NS model, and its rhabdom diameter equals the rhabdomere diameter in fly $d_{rh} = 1.9$ µm. The simple eye is hemispherical (**Figure 7a**), therefore the specific volume of optics is given by

$$V_o = \frac{\pi}{3}f^3 = \frac{\pi}{3}(DF)^3 \, \mu\text{m}^3 \, \text{sr}^{-1} , \tag{45}$$

and with a photoreceptor array with rhabdoms of length $L$, the total specific volume of the eye is given by

$$V_{tot} = \frac{\pi}{3}(DF + L)^3 , \tag{46}$$

and the specific volume of the photoreceptor array by

$$V_{ph} = V_{tot} - V_o . \tag{47}$$

The receptor spacing, $\Delta\phi$, depends on the lens focal length, $f$ and $d_{rh}$,

$$\Delta\phi = \frac{d_{rh}}{f} = \frac{d_{rh}}{DF} . \tag{48}$$

This is an important difference between simple eyes and apposition eyes. In apposition eyes, the model $\Delta\phi$ is can be varied at constant $f$ and $D$ by changing the optical radius $R$. However, in a simple eye with a diffraction limited lens, the relationship between the optical resolving power of the lens and the anatomical resolving power of the photoreceptor mosaic depends on $F$-number, $F$, as follows (**Snyder, 1979**). For a hexagonal mosaic of photoreceptors, the diffraction limit is reached when

$$\lambda/D = \sqrt{3}\frac{d_{rh}}{f} = \sqrt{3}\frac{d_{rh}}{DF}. \tag{49}$$

When $\lambda$, the wavelength of light is $500$ nm, and $d_{rh} = 2$ µm, then $F = 6.9$. It follows that our model simple eye with $F = 2$ is undersampling by a factor of approximately 3.5. Running our simple eye model with $F = 4$ had very little effect on the optimum $L$ (data not shown), suggesting that our conclusion that efficient simple eyes have much shorter rhabdoms than efficient apposition eyes is robust with respect to undersampling.

### Calculating $N_{vil}$ as a function of $L$

We estimate $v$, the number of microvilli per unit length of rhabdomere, for our NS eye model using findings from photoreceptors R1–6 in female blowflies. Dividing the number of microvilli in a rhabdomere, $6 \times 10^4$ (*Howard et al., 1987*), by $L$ = 260 μm, which is towards the upper end of the range $L$ = 220 to 280 μm reported for blowfly (*Hardie, 1985*), gives $v$ = 230 μm$^{-1}$. Although fly rhabdomeres vary in both diameter and degree of taper according to eye region, sex and species, and this sculpting adapts photoreceptor arrays to visual ecology, for example (*Gonzalez-Bellido et al., 2011*), we fix $v$ to simplify the derivation of points of principle.

For our generic model of a fused rhabdom apposition eye, we consider a column of six photoreceptors that, like the short visual fibres R1–6 in fly, terminate in the first optic neuropile. Their six rhabdomeres form the triradiate rhabdom described in locust (*Wilson et al., 1978*). This fused rhabdom has a cross-section that approximates an equilateral triangle in which each side is constructed by a pair of photoreceptors whose parallel microvilli are equivalent to a single fly R1–6 rhabdomere. Thus, each pixel is coded by a set of six rhabdomeres that are equivalent to three fly R1–6 rhabdomeres. Therefore, the number of transduction units coding a pixel is half that in fly and the resulting changes in signal and noise are simply accounted for by replacing the factor $(3\psi)^2$ in *Equation 29* with $\left(\frac{3\psi}{2}\right)^2$, and $3\psi$ in *Equations 33 and 34* with $\left(\frac{3\psi}{2}\right)$.

For our generic model of a diurnal simple eye, we assume that, as in spiders, the rhabdomeres of adjacent photoreceptors abut; that is there are no gaps between photoreceptors; therefore, a photoreceptor's rhabdomeres project from a central column of cytoplasm. To compare with our apposition models, we assume that the diameter of the photoreceptor entrance aperture, $d_{rh}$ = 1.9 μm and contains microvilli projecting in three directions. In this case, the simple eye photoreceptor is equivalent to the triradiate fused rhabdom.

## Acknowledgements

We are grateful to Doekele Stavenga for his expert advice on optics and providing the micrograph in *Figure 1*, to Jeremy Niven for his advice during the project and his comments on the manuscript, and to Daniel Osorio and James Herbert-Read for their comments on the manuscript.

## Additional information

### Funding

| Funder | Grant reference number | Author |
| --- | --- | --- |
| La Caixa Postgraduate Scholarship | | Francisco JH Heras |
| Caja-Madrid Postgraduate Scholarship | | Francisco JH Heras |
| H. Britton Sanderford Honorarium | | Simon B Laughlin |

The funders had no role in study design, data collection, and interpretation, or the decision to submit the work for publication.

### Author contributions

Francisco JH Heras, Conceptualization, Data curation, Software, Formal analysis, Funding acquisition, Investigation, Methodology, Writing – original draft, Writing – review and editing; Simon B Laughlin, Conceptualization, Data curation, Supervision, Investigation, Visualization, Methodology, Writing – original draft, Writing – review and editing

### Author ORCIDs

Francisco JH Heras (iD) https://orcid.org/0000-0001-8124-2359
Simon B Laughlin (iD) https://orcid.org/0000-0003-4659-6543

Reviewer #1 (Public review): https://doi.org/10.7554/eLife.96517.3.sa1
Reviewer #2 (Public review): https://doi.org/10.7554/eLife.96517.3.sa2
Reviewer #3 (Public review): https://doi.org/10.7554/eLife.96517.3.sa3
Author response https://doi.org/10.7554/eLife.96517.3.sa4

## Additional files

### Supplementary files

Supplementary file 1. Annotated spreadsheet of the anatomical and optical data extracted from the literature that was used to provide the values given in *Tables 2 and 3*, and plotted in *Figures 5 and 6*.

MDAR checklist

### Data availability

The code for the models used in this paper is openly accessible at https://github.com/fjhheras/eyede-sign (copy archived at *Heras, 2025*). The files *Figure 4—source data 1*, *Figure 5—source data 1*, *Figure 6—source data 1*, *Figure 7—source data 1*, *Figure 8—source data 1* contain the numerical data used to generate the graphs in these figures.

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
