## [Editor Report · eLife Assessment]

This paper makes a **valuable** contribution to our understanding of the tradeoffs in eye design - specifically between improvements in optics and in photoreceptor performance. The authors successfully build a formal theory that enables comparisons across a wide range of species and eye types. One notable example is that how space should be allocated to optics and photoreceptors depends on eye type - with particularly notable differences between compound and simple eyes. The framework introduced to compare different design properties is **convincing** and provides a nice example of how to study tradeoffs in seemingly disparate design properties.

---

## [Referee Report · Reviewer #1 (Public review)]

Summary:

Two important factors in visual performance are the resolving power of the lens and the signal-to-noise ratio of the photoreceptors. These both compete for space: a larger lens has improved resolving power over a smaller one, and longer photoreceptors capture more photons and hence generate responses with lower noise. The current paper explores the tradeoff of these two factors, asking how space should be allocated to maximize eye performance (measured as encoded information).

The revisions, to my read, have greatly improved the paper. Most of this was due to setting clear expectations from the start of the paper. Nice work!

---

## [Referee Report · Reviewer #2 (Public review)]

Summary:

In short, the paper presents a theoretical framework that predicts how resources should be optimally distributed between receptors and optics in eyes.

After revision of an already excellent contribution, the manuscript is now even better. The authors have responded carefully to all reviewer comments.

Strengths:

The authors build on the principle of resource allocation within an organism and develop a formal theory for optimal distribution of resources within an eye between the receptor array and the optics. Because the two parts of eyes, receptor arrays and optics, share the same role of providing visual information to the animal it is possible to isolate these from resource allocation in the rest of the animal. This allows for a novel and powerful way of exploring the principles that govern eye design. By clever and thoughtful assumptions/constraints, the authors have built a formal theory of resource allocation between the receptor array and the optics for two major types of compound eye as well as for camera-type eyes. The theory is formalized with variables that are well characterized in a number of different animal eyes, resulting in testable predictions.

The authors use the theory to explain a number of design features that depend on different optimal distribution of resources between the receptor array and the optics in different types of eye. As an example, they successfully explain why eye regions with different spatial resolution should be built in different ways. They also explain differences between different types of eye, such as long photoreceptors in apposition compound eyes and much shorter receptors in camera type eyes. The predictive power in the theory is impressive.

To keep the number of parameters at a minimum, the theory was developed for two types of compound eye (neural superposition, and apposition) and for camera-type eyes. It is possible to extend the theory to other types of eye, although it would likely require more variables and assumptions/constraints to the theory. It is thus good to introduce the conceptual ideas without overdoing the applications of the theory.

The paper extends a previous theory, developed by the senior author, that develops performance surfaces for optimal cost/benefit design of eyes. By combining this with resource allocation between receptors and optics, the theoretical understanding of eye design takes a major leap and provides entirely new sets of predictions and explanations for why eyes are built the way they are.

The paper is well written and even though the theory development in the Results may be difficult to take in for many biologists, the Discussion very nicely lists all the major predictions under separate headings, and here the text is more tuned for readers that are not entirely comfortable with the formalism of the Results section. I must point out though that the Results section is kept exemplary concise. The figures are excellent and help explain concepts that otherwise may go above the head of many biologists.

---

## [Referee Report · Reviewer #3 (Public review)]

Summary:

This is a proposal for a new theory for the geometry of insect eyes. The novel cost-benefit function combines the cost of the optical portion with the photoreceptor portion of the eye. These quantities are put on the same footing using a specific (normalized) volume measure, plus an energy factor for the photoreceptor compartment. An optimal information transmission rate then specifies each parameter and resource allocation ratio for a variable total cost. The elegant treatment allows for comparison across a wide range of species and eye types. Simple eyes are found to be several times more efficient across a range of eye parameters than neural superposition eyes. Some trends in eye parameters can be explained by optimal allocation of resources between the optics and photoreceptors compartments of the eye.

Strengths:

Data from a variety of species roughly align with rough trends in the cost analysis, e.g. as a function of expanding the length of the photoreceptor compartment.

New data could be added to the framework once collected, and many species can be compared.

Eyes of different shapes are compared.

Weaknesses:

Detailed quantitative conclusions are not possible given the approximations and simplifying assumptions in the models and weak accounting for trends in the data across eye types.

Comments on revisions:

I have no additional comments for the authors and appreciate the revisions and corrections implemented - I think those changes have improved the clarity of the manuscript and expanded the potential readership for the paper.

---

## [Author Response]

The following is the authors’ response to the original reviews.

**Reviewer #1 (Public Review):**
Summary:Two important factors in visual performance are the resolving power of the lens and the signal-to-noise ratio of the photoreceptors. These both compete for space: a larger lens has improved resolving power over a smaller one, and longer photoreceptors capture more photons and hence generate responses with lower noise. The current paper explores the tradeoff of these two factors, asking how space should be allocated to maximize eye performance (measured as encoded information).Your summary is clear, concise and elegant. The competition is not just for space, it is for space, materials and energy. We now emphasise that we are considering these three costs in our rewrites of the Abstract and the first paragraph of the Discussion.Strengths:The topic of the paper is interesting and not well studied. The approach is clearly described and seems appropriate (with a few exceptions - see weaknesses below). In most cases, the parameter space of the models are well explored and tradeoffs are clear.Weaknesses:Light levelThe calculations in the paper assume high light levels (which reduces the number of parameters that need to be considered). The impact of this assumption is not clear. A concern is that the optimization may be quite different at lower light levels. Such a dependence on light level could explain why the model predictions and experiment are not in particularly good agreement. The paper would benefit from exploring this issue.

Thank you for raising this point. We briefly explained in our original Discussion, under Understanding the adaptive radiation of eyes (Version 1, Iines 756 – 762), how our method can be modified to investigate eyes adapted for lower light levels. We have some thoughts on how eyes might be adapted. In general, transduction rates are increased by increasing *D*, reducing *f*, increasing *d_rh_* and increasing *L* . In addition, *d*_*rh*_ is increased to allow for a larger *D* within the constraint of eye radius/corneal surface area, and to avoid wasteful oversampling (the changes in *D, f* and *d_rh_* increase acceptance angle ∆*ρ*). We suspect that in eyes optimised for the efficient use of space, materials and energy the increases in *L* will be relatively small, first because increasing *D*, reducing *f* and increasing *drh* are much more effective at increasing transduction rate than increasing *L*. Second, increasing sensitivity by reducing *f* decreases the cost *V_o_* whereas increasing sensitivity by increasing *L* increases the cost V_ph_. This disadvantage, together with exponential absorption, might explain why *L* is only 10% - 20% longer in the apposition eyes of nocturnal bees (Somanathan et al, J. comp. Physiol. A195, 571583, 2009). Because this line of argument is speculative and enters new territory, we have not included it in our revised version. We already present a lot of new material for readers to digest, and we agree with referee 2 that “It is possible to extend the theory to other types of eyes, although it would likely require more variables and assumptions/constraints to the theory. It is thus good to introduce the conceptual ideas without overdoing the applications of the theory”. Nonetheless, we take your point that some of the eyes in our data set might be adapted for lower light levels, and we have rewritten the Discussion section, How efficiently do insects allocate resources within their apposition eyes accordingly. On line 827 – 843 we address the assumption that eyes are adapted for full daylight, and also take the opportunity to mention two more reasons for increasing the eye parameter *p*: namely increasing image velocity (Snyder, 1979), and constructing bright zones that increase the detectability of small targets (van Hateren et al., 1989; Straw et al., 2006).

DiscontinuitiesThe discontinuities and non-monotonicity of the optimal parameters plotted in Figure 4 are concerning. Are these a numerical artifact? Some discussion of their origin would be quite helpful.

Good points, we now address the discontinuities in the Results, where they are first observed (lines 311 - 319)

Discrepancies between predictions and experimentAs the authors clearly describe, experimental measurements of eye parameters differ systematically from those predicted. This makes it difficult to know what to take away from the paper. The qualitative arguments about how resources should be allocated are pretty general, and the full model seems a complex way to arrive at those arguments. Could this reflect a failure of one of the assumptions that the model rests on - e.g. high light levels, or that the cost of space for photoreceptors and optics is similar? Given these discrepancies between model and experiment, it is also hard to evaluate conclusions about the competition between optics and photoreceptors (e.g. at the end of the abstract) and about the importance for evolution (end of introduction).

Your misgivings boil down to two issues: what use is a model that fails to fit the data, and do we need a complicated model to show something that seems to be intuitively obvious? Our study is useful because it introduces new approaches, methods, factors and explanations which advance our analysis and understanding of eye design and evolution. Your comments make it clear that we failed to get this message across and we have revised the manuscript accordingly. We have rewritten the Abstract and the first paragraph of the Discussion to emphasise the value of our new measure of cost, specific volume, by including more of its practical advantages. In particular, our use of specific volume (1) opens the door to the morphospace of all eyes of given type and cost. (2) This allows one to construct performance surfaces across morphospace that not only identify optima, but by evaluating the sub-optimal cast light on efficiency and adaptability. (3) Shows that photoreceptor energy costs have a major impact on design and efficiency, and (4) allows us to calculate and compare the capacities and efficiencies of compound eyes and simple eyes using a superior measure of cost. It is also possible that your dissatisfaction was deepened by disappointment. The first sentence of our original Abstract said that the goal of design is to maximize performance, so you might have expected to see that eyes are optimised. Given that optimization provides cast iron proof that a system is designed to be efficient, and previous studies of coding by fly LMCs (Laughlin, 1981; Srinivasan et al., 1982 & van Hateren 1992) validated Barlow’s Efficient Coding Hypothesis by showing that coding is optimised, your expectation is reasonable. However, our investigation of how the allocation of resources to optics and photoreceptors affects an eye’s performance, efficiency and design does not depend *a priori* on finding optima, therefore we have removed the “maximized”. Our revised Abstract now says, “to improve performance”.

In short, our study illustrates an old adage in statistics “All models fail to fit, but some are useful”. As is often the case, the way in which our model fails is useful. In the original version of the Results and Discussion, we argued that the allocation of resources is efficient, and identified factors that can, in principle, explain the scattering of data points. Indeed, our modelling identifies two of these deficiencies; a lack of data on species-specific energy usage, and the need for models that quantify the relationship between the quality of the captured image and the behavioural tasks for which an eye might be specialised. Thus, by examining the model’s failings we identify critical factors and pose new questions for future research. We have rewritten the Discussion section How efficiently do insects allocate resources**….** to make these points. We hope that these revisions will convince you that we have established a starting point for definitive studies, invented a vehicle that has travelled far enough to discover new territory, and shown that it can be modified to cope with difficult terrain.

Turning to the need for a complicated model, because the costs and benefits depend on elementary optics and geometry, we too thought that there ought to be a simple model. However, when we tried to formulate a simple set of equations that approximate the definitive findings of our more complicated model we discovered that this is not as straightforward as we thought. Many of the parameters in our model interact to determine costs and benefits, and many of these interactions are non-linear (e.g. the volumes of shells in spheres involve quadratic and cubic terms, and information depends on the log of a square root). So, rather than hold back publication of our complicated model, we decided to explain how it works as clearly as we can and demonstrate its value.

In response to your final comment, “it is hard to evaluate conclusions about the competition between optics and photoreceptors (e.g. at the end of the abstract) and about the importance for evolution (end of introduction)”, we stand by our original argument. There must be competition in an eye of fixed cost, and because competition favours a heavy investment in photoreceptors, both in theory and in practice, it is a significant factor in eye design. A match between investments in optics and photoreceptors is predicted by theory and observed in fly NS eyes, therefore this is a design principle. As for evolution, no one would deny that it is important to view the adaptive radiation of eyes through a cost-benefit lens. Our lens is the first to view the whole eye, optics and photoreceptor array, and the first to treat the costs of space, materials and energy. Although the view through our lens is a bit fuzzy, it reveals that costs, benefits and trade-offs are important. Thus we have established a promising starting point for a new and more comprehensive cost-benefit approach to understanding eye design and evolution. As for the involvement of genes, when there are heritable changes in phenotype genes must be involved and if, as we suggest, efficient resource allocation is beneficial, the developmental mechanisms responsible for allocating resources to optics and photoreceptor array will be playing a formative role in eye evolution.

**Reviewer #2 (Public Review):**
Summary:In short, the paper presents a theoretical framework that predicts how resources should be optimally distributed between receptors and optics in eyes.Strengths:The authors build on the principle of resource allocation within an organism and develop a formal theory for optimal distribution of resources within an eye between the receptor array and the optics. Because the two parts of eyes, receptor arrays and optics, share the same role of providing visual information to the animal it is possible to isolate these from resource allocation in the rest of the animal. This allows for a novel and powerful way of exploring the principles that govern eye design. By clever and thoughtful assumptions/constraints, the authors have built a formal theory of resource allocation between the receptor array and the optics for two major types of compound eye as well as for camera-type eyes. The theory is formalized with variables that are well characterized in a number of different animal eyes, resulting in testable predictions.The authors use the theory to explain a number of design features that depend on different optimal distribution of resources between the receptor array and the optics in different types of eyes. As an example, they successfully explain why eye regions with different spatial resolution should be built in different ways. They also explain differences between different types of eyes, such as long photoreceptors in apposition compound eyes and much shorter receptors in camera type eyes. The predictive power in the theory is impressive.To keep the number of parameters at a minimum, the theory was developed for two types of compound eye (neural superposition, and apposition) and for camera-type eyes. It is possible to extend the theory to other types of eyes, although it would likely require more variables and assumptions/constraints to the theory. It is thus good to introduce the conceptual ideas without overdoing the applications of the theory.The paper extends a previous theory, developed by the senior author, that develops performance surfaces for optimal cost/benefit design of eyes. By combining this with resource allocation between receptors and optics, the theoretical understanding of eye design takes a major leap and provides entirely new sets of predictions and explanations for why eyes are built the way they are.The paper is well written and even though the theory development in the Results may be difficult to take in for many biologists, the Discussion very nicely lists all the major predictions under separate headings, and here the text is more tuned for readers that are not entirely comfortable with the formalism of the Results section. I must point out though that the Results section is kept exemplary concise. The figures are excellent and help explain concepts that otherwise may go above the head of many biologists.

We are heartened by your appreciation of our manuscript - it persuaded us not to undertake extensive revisions – thank you.

**Reviewer #3 (Public Review):**
Summary:This is a proposal for a new theory for the geometry of insect eyes. The novel costbenefit function combines the cost of the optical portion with the photoreceptor portion of the eye. These quantities are put on the same footing using a specific (normalized) volume measure, plus an energy factor for the photoreceptor compartment. An optimal information transmission rate then specifies each parameter and resource allocation ratio for a variable total cost. The elegant treatment allows for comparison across a wide range of species and eye types. Simple eyes are found to be several times more efficient across a range of eye parameters than neural superposition eyes. Some trends in eye parameters can be explained by optimal allocation of resources between the optics and photoreceptors compartments of the eye.Strengths:Data from a variety of species roughly align with rough trends in the cost analysis, e.g. as a function of expanding the length of the photoreceptor compartment.New data could be added to the framework once collected, and many species can be compared.Eyes of different shapes are compared.Weaknesses:Detailed quantitative conclusions are not possible given the approximations and simplifying assumptions in the models and poor accounting for trends in the data across eye types.
**Reviewer #1 (Recommendations For The Authors):**
Figure 1: Panel E defines the parameters described in panel d. Consider swapping the order of those panels (or defining D and Delta Phi in the figure legend for d). Order follows narrative, eye types then match

We think that you are referring to Figure 1. We modified the legend.

Lines 143-145: How does a different relative cost impact your results?

Thank you for raising this question. Because our assumption that relative costs are the same is our starting point, and for optics it is not an obvious mistake, we do not raise your question here. We address your question where you next raise it because, for photoreceptors the assumption is obviously wrong. We now emphasise that our method for accounting for photoreceptor energy costs can be applied to other costs.

Lines 187-190: Same as above - how do your results change if this assumption is not accurate?

We have revised our manuscript to emphasise that we are dealing with the situation in which our initial assumption (costs per unit volume are equal) breaks down. On (lines 203 - 208) we write “ However, this assumption breaks down when we consider specific metabolic rates. To enable and power phototransduction, photoreceptors have an exceptionally high specific metabolic rate (energy consumed per gram, and hence unit volume, per second) (Laughlin et al., 1998; Niven et al., 2007; Pangršič et al., 2005). We account for this extra cost by applying an energy surcharge, S_E_. To equate….

We also revised part of the Discussion section**,** Specific volume is a useful measure of cost, to make it clear that we are able take account for situations in which the costs per unit volume are not equal, and we give our treatment of photoreceptor energy costs as an example of how this is done. On lines 626 - 640 we say

Cost estimates can be adjusted for situations in which costs per unit volume are not equal, as illustratedby our treatment of photoreceptor energy consumption. To support transduction the photoreceptor array has an exceptionally high metabolic rate (Laughlin et al., 1998; Niven et al., 2007; Pangršič et al., 2005). We account forthis higher energy cost by using the animal’s specific metabolic rate (power per unit mass and hence power per unit volume) to convert an array’s power consumption into an equivalent volume (Methods). Photoreceptor ion pumps are the major consumers of energy and the smaller contribution of pigmented glia (Coles, 1989) is included in our calculation of the energy tariff *KE*. (Methods) The higher costs of materials and their turnover in the photoreceptor array can be added the energy tariff *KE* but given the magnitude of the light-gated current (Laughlin et al., 1998) the relative increase will be very small. Thus for our intents and purposes the effects of these additional costs are covered by our models. For want of sufficient data…”.

**Reviewer #2 (Recommendations For The Authors):**

A few comments for consideration by the authors:

(1) In the abstract, Maybe give another example explaining why other eyes should be different to those of fast diurnal insects.

This worthwhile extrapolation is best kept to the Discussion.

(2) Would it be worthwhile mentioning that the photopigment density is low in rhabdoms compared to vertebrate outer segments? This will have major effects on the relative size of retina and optics.

Thank you, we now make this good point in the Discussion (lines 698-702).

(3) It took me a while to understand what you mean by an energy tariff. For the less initiated reader many other variables may be difficult to comprehend. A possible remedy would be to make a table with all variables explained first very briefly in a formal way and then explained again with a few more words for readers less fluent in the formalism.

A very useful suggestion. We have taken your advice (p.4).

(4) The "easy explanation" on lines 356-357 need a few more words to be understandable.

We have expanded this argument, and corrected a mistake, the width of the head front to back is not 250 μm, it is 600 μm (lines 402-407)

(5) Maybe devote a short paragraph in the Discussion to other types of eye, such as optical superposition eyes and pinhole eyes. This could be done very shortly and without formalism. I'm sure the authors already have a good idea of the optimal ratio of receptor arrays and optics in these eye types.

We do not discuss this because we have not found a full account of the trade-offs and their effects on costs and benefits. We hope that our analysis of apposition and simple eyes will encourage people to analyse the relationships between costs and benefits in other eye types. To this end we pointed out in the Discussion that recent advances in imaging and modelling could be helpful.

(6) Could the sentence on lines 668-671 be made a little clearer?“Efficiency is also depressed by increasing the photoreceptor energy tariff *KE*, and in line with the greater impact of photoreceptor energy costs in simple eyes, the reduction in efficiency is much greater in simple eyes (Figure 8b).0.

We replaced this sentence with “In both simple and apposition eyes efficiency is reduced by increasing the photoreceptor energy tariff *KE*. This effect is much greater in simple eyes, thus as found for reductions in photoreceptor length (Figure 7b),*KE* has more impact on the design of simple eyes” (lines749 – 752).

(7) I have some reservations about the text on lines 789-796. The problem is that optics can do very little to improve the performance of a directional photoreceptor where delrho should optimally be very wide. Here, membrane folding is the only efficient way to improve performance (SNR). The option to reduce delrho for better performance comes later when simultaneous spatial resolution (multiple pixels) is introduced.

Yes, we have been careless. We have rewritten this paragraph to say (lines 920-931)

“Two key steps in the evolution of eyes were the stacking of photoreceptive membranes to absorb more photons, and the formation of optics to intercept more photons and concentrate them according to angle of incidence to form an image (Nilsson, 2013, 2021). Our modelling of well-developed image forming eyes shows that to improve performance stacked membranes (rhabdomeres) compete with optics for the resources invested in an eye, and this competition profoundly influences both form and function. It is likely that competition between optics and photoreceptors was shaping eyes as lenses evolved to support low resolution spatial vision. Thus the developmental mechanisms that allocate resources within modern high resolution eyes (Casares & MacGregor, 2021), by controlling cell size and shape, and as our study emphasises, gradients in size and shape across an eye, will have analogues or homologues in more ancient eyes. Their discovery….” (lines 920-931)

**Reviewer #3 (Recommendations For The Authors):**
Suggestions for major revisions:While the approach is novel and elegant, the results from the analysis of insect morphology do not broadly support the optimization argument and hardly constrain parameters, like the energy tariff value, at all. The most striking result of the paper is the flat plateau in information across a broad range of shape parameters and the length, and resolution trend in Figure 5.

At no point in the Results and Discussion do we argue that resource allocation is optimized. Indeed, we frequently observe that it is not. Our mistake was to start the Abstract by observing that animals evolve to minimise costs. We have rewritten the Abstract accordingly.

The information peaks are quite shallow. This might actually be a very important and interesting result in the paper - the fact that the information plateaus could give the insect eye quite a wide range of parameters to slide between while achieving relatively efficient sensing of the environment. Instead of attempting to use a rather ad hoc and poorly supported measure of energetics in PR cost, perhaps the pitch could focus on this flexibility. *KE* does not seem to constrain eye parameters and does not add much to the paper.

We agree, being able to construct performance surfaces across morphospace is an important advance in the field of eye design and evolution, and the performance surface’s flat top has interesting implications for the evolution of adaptations. Encouraged by your remarks, we have rewritten the Abstract and the introductory paragraph of the Discussion to draw attention to these points.

We are disappointed that we failed to convince you that our energy tariff, *K_E_* , is no better than a poorly supported ad hoc parameter that does not add much to the paper. In our opinion a resource allocation model that ignores photoreceptor energy consumption is obviously inadequate because the high energy cost of phototransduction is both wellknown and considered to be a formative factor in eye evolution (Niven and Laughlin, 2008). One of the advantages of modelling is that one can assess the impact of factors that are known to be present, are thought to be important, but have not been quantified. We followed standard modelling practice by introducing a cost that has the same units as the other costs and, for good physiological reasons, increases linearly with the number of microvilli, according to *K_E_*. We then vary this unknown cost parameter to discover when and why it is significant. We were pleased to discover that we could combine data on photoreceptor energy demands and whole animal metabolic rates to establish the likely range of *K_E_*. This procedure enabled us to unify the cost-benefit analyses of optics and photoreceptors, and to discover that realistic values of *K_E_* have a profound impact on the structure and performance of an efficient eye. We hope that this advance will encourage people to collect the data needed to evaluate *K_E_*.To emphasise the importance of *K_E_* and dispel doubts associated with the failure of the model to fit the data, we have revised two sections: Flies invest efficiently in costly photoreceptor arrays in the Results, and How efficiently do insects allocate resources within their apposition eyes? in the Discussion. These rewrites also explain why it is impossible for us to infer *K_E_* by adjusting its value so that the model’s predictions fit the data.

The graphics after Figure 3 are quite dense and hard to follow. None of the plateau extent shown in Fig 3 is carried through to the subsequent plots, which makes the conclusions drawn from these figures very hard to parse. If the peak information occurs on a flat plateau, it would be more helpful to see those ranges of parameters displayed in the figures.Ideally one should do as you suggest and plot the extent of the plateau, but in our situation this is not very helpful. In the best data set, flies, optimised models predict D well, get close to ∆φ in larger eyes, and demonstrate that these optimum values are not very sensitive to *KE. L* is a different matter, it is very sensitive to *KE* which, as we show (and frequently remind) is poorly constrained by experimental data. The best we can do is estimate the envelope of *L* vs *C_tot_* curves, as defined by a plausible range of *KEL* . Because most of the plateau boundaries you ask for will fall within this envelope, plotting them does little to clear the fog of uncertainty. We note that all three referees agree that our model can account for two robust trends, (i) in apposition eyes L increase with optical resolving power and acuity, both within individual eyes and among eyes of different sizes, and (ii) L is much longer is apposition eyes than in simple eyes. Nonetheless, the scatter of data points and their failure to fit creates a bad impression. We gave a number of reasons why the model does not fit the data points, but these were scattered throughout the Results and Discussion and, as referees 1 and 3 point out, this makes it difficult to draw convincing conclusions. To rectify this failing, we have rewritten two sections, in the Results Flies invest efficiently in costly photoreceptor arrays and in the Discussion, How efficiently do insects allocate resources within their apposition eyes?, to discuss these reasons en bloc, draw conclusions and suggest how better data and refinements to modelling could resolve these issues.Throughout the figures, the discontinuities in the optimal cuts through parameter space are not sufficiently explained.

We added a couple of sentences that address the “jumps” (lines 313 – 318)

None of the data seems to hug any of the optimal lines and only weakly follow the trends shown in the plots. This makes interpretation difficult for the reader and should be better explained. The text can be a little telegraphic in the Results after roughly page 10, and requires several readings to glean insight into the manuscript's conclusions.

We revised the Results section in which we compare the best data set, flies’ NS eyes with theoretical predictions, Flies invest efficiently in costly photoreceptor arrays, to expand our interpretation of the data and clarify our arguments. The remaining sections have not been expanded. In the next section, which is on fused rhabdom apposition eyes, our interpretation of the scattering of data points follows the same line of argument. The remaining Results sections are entirely theoretical.

Overall, the rough conclusions outlined in the Results seem moderately supported by the matches of the data to the optimal information transmission cuts through parameter space, but only weakly.

We agree, more data is required to test and refine our theoretical predictions.

The Discussion is long and well-argued, and contains the most cogent writing in the manuscript.

Thank you: this is most pleasing. We submitted our study to eLife because it allows longer Discussions, but we worried that ours was too long. However, we felt that our extensive Discussion was necessary for two reasons. First, we are introducing a new approach to understanding of eye design and evolution. Second, because the data on eye morphology and costs are limited, we had to make a number of assumptions and by discussing these, warts and all, we hoped to encourage experimentalists to gather more data and focus their efforts on the most revealing material.

Minor comments:

We have acted upon most of your minor comments and we confine our remarks to our disagreements. We are grateful for your attention to details that we \textshould have picked up on.

It's a more standard convention to say "cost-benefit" rather than with a colon."equation" should be abbreviated "eq" or "eqn", never with a "t"when referring to the work of van Hateren, quote the paper and the database using "van Hateren" not just "Hateren"small latex note: use "\textit{SNR}" to get the proper formatting for those letters when in the math environmentLine 100-110: "f" is introduced, but only f' is referenced in the figure. This should be explained in order. d_rh is not included in the figure. Also in this section, d_rh/f is also referenced before \Delta \rho_rf, which is the same quantity, without explanation.

Figure 1 shows eye structure and geometry. *f’* is a lineal dimension of the eye but *f* is not, so *f* is not shown in Fig 1e. We eliminated the confusion surrounding *∆ρ*_*rh*_ by deleting “and changing the acceptance angle of the photoreceptive waveguide ∆ρ_rh_ (Snyder, 1979)”.

Fig 1 caption: this says "From dorsal to ventral," then describes trends that run ventral to dorsal, which is a confusing typo.Fig 3 - adding some data points to these plots might help the reader understand how (or if) K_E is constrained by the data.

It is not possible to add data points because to total cost, *C_tot_* ,is unknown.

Fig 4c (and in other subplots): the jumps in L with C_tot could be explained better in the text - it wasn't clear to this reviewer why there are these discontinuities.

Dealt with in the revised text (lines 310-318).

Fig 4d: The caption for this subplot could be more clearly written.

We have rewritten the subscript for subplot 4d.

Fig 5 and other plots with data: please indicate which symbols are samples from the same species. This info is hard to reconstruct from the tables.

We have revised Figure 5 accordingly. Species were already indicated in Figure 6.

Line 328: missing equation number